# Neolithic culinary traditions revealed by cereal, milk and meat lipids in pottery from Scottish crannogs

Simon Hammann [1,2] ✉, Rosie R. Bishop[3], Mike Copper[4], Duncan Garrow[5] ✉, Caitlin Greenwood[1], Lanah Hewson[5,9], Alison Sheridan [6], Fraser Sturt[7] ✉, Helen L. Whelton [8] & Lucy J. E. Cramp [1] ✉

Cereal cultivation in Britain dates back to ca. 4000 BCE, probably introduced by migrant farmers from continental Europe. Widespread evidence for livestock appears in the archaeozoological record, also reflected by ubiquitous dairy lipids in pottery organic residues. However, despite archaeobotanical evidence for domesticated plants (such as cereals), organic residue evidence has been near-absent. Our approach, targeting low-abundance cereal-specific markers, has now revealed evidence for cereals (indicating wheat) in Neolithic pottery from Scottish 'crannogs', dating to ca. 3600 – 3300 BCE. Their association with dairy products suggests cereals may have been regularly prepared together as a milk-based gruel. We also observed a strong association between the occurrence of dairy products and smaller-mouthed vessels. Here, we demonstrate that cereal-specific markers can survive in cooking pots for millennia, revealing the consumption of specific cereals (wheat) that are virtually absent from the archaeobotanical record for this region and illuminating culinary traditions among early farming communities.

The consumption of domesticated plants and animals first emerged in Britain and Ireland in the centuries around 4000 BCE, and it accompanies other novel traditions, practices and technology, including the use of pottery and new settlement and funerary practices[1]. Recent analysis of ancient DNA confirms the view that migrant farmers from continental Europe were responsible for introducing these new practices[2] and there is widespread evidence for the prevalence of domesticated animals and dairying amongst Britain and Ireland's first farmers, even at the outermost locations of the archipelago[3]. Archaeobotanical evidence suggests that cereals were also consumed by Britain and Ireland's first farmers. However, there may have been considerable regional variation in the importance of cereals within different culinary traditions across Britain and Ireland[4–8]. Cereal grains are consistently present in Neolithic archaeobotanical assemblages from across Britain and Ireland, though often in relatively small numbers (<500 grains)[4–8], and rarely in direct association with pottery vessels or only in ceremonial contexts, precluding the identification of specific culinary practices.

Here we apply a new method to assess directly the presence and nature of the use of cereals in Neolithic culinary practices. By using a

[1]Department of Anthropology and Archaeology, University of Bristol, 43 Woodland Road, Bristol BS81UU, UK. [2]Department of Chemistry and Pharmacy, Friedrich-Alexander-Universität Erlangen-Nürnberg, Nikolaus-Fiebiger Straße 10, 91058 Erlangen, Germany. [3]Arkeologisk Museum, Universitetet i Stavanger, Peder Klows gate 31A, 4036 Stavanger, Norway. [4]School of Archaeological and Forensic Sciences, University of Bradford, Richmond Road, Bradford BD7 1DP, UK. [5]Department of Archaeology, University of Reading, Whiteknights Box 227, Reading RG6 6AB, UK. [6]c/o Scottish History & Archaeology Department, National Museums Scotland, Chambers St, Edinburgh EH1 1JF, UK. [7]Department of Archaeology, University of Southampton, Avenue Campus, Highfield, Southampton SO17 1BF, UK. [8]Organic Geochemistry Unit, School of Chemistry, University of Bristol, Cantock's Close, Bristol BS8 1TS, UK. [9]Present address: Museum of London Archaeology (MOLA) Northampton, 30 Billing Road, Northampton NN1 5DQ, UK. ✉e-mail: simon.hammann@fau.de; d.j.garrow@reading.ac.uk; F.Sturt@soton.ac.uk; lucy.cramp@bristol.ac.uk

highly sensitive approach to analyse organic residues extracted from Neolithic pottery, we are able to directly detect specific molecular biomarkers of the cereals that were cooked in the vessels themselves. We focused on pottery recovered from a group of artificial/semi-artificial islands known as 'crannogs'. The exceptional contextual conditions of artefacts from four recently discovered sites raised the possibility that prehistoric cereal processing and/or cooking in pots may be detected through the recovery of cereal-specific biomarkers that we have previously shown to be likely to survive only under anoxic conditions[9]. This approach also opens up the possibility of exploring how a fundamental element of the Neolithic diet was being prepared and consumed, while simultaneously shedding new light on the lives and habits of the people that inhabited or visited these little-understood crannog sites.

Crannogs are an intriguing and puzzling category of archaeological site. These artificial constructions in lakes, including dwellings, occur throughout prehistory, in many historic periods and across the contemporary world. Such constructions are often domestic but can have other functions, such as providing locations for ritual practice. Despite the widespread geographical and temporal distribution of crannogs, the human activities associated with islet sites can be hard to discern[10–12]. Certainly, many of the more recent sites, such as the Iron Age islet 'duns' of the Outer Hebrides, are likely to have been used for defensive purposes. However, a defensive function is not necessarily apparent for many of those sites with prehistoric origins. Some were built in very shallow waters close to the shore, while others appear too small to have housed significant structures that would denote long-term occupation[10,12]. Consequently, a number of archaeologists have suggested that they may have been built out on the water for symbolic reasons – to express a group's social separation from the rest of society or to create a special ritual space separate from everyday life[13–15].

Recent excavations and underwater surveys at four crannogs on the Isle of Lewis in the Outer Hebrides produced substantial quantities of artefactual material, including Neolithic pottery (Fig. 1)[15–17]. In addition to the style of pottery, direct radiocarbon dates from burnt-on residues and from structural timbers ranging from 3640 cal BCE to 3360 cal BCE demonstrate that these sites were occupied during the Neolithic (Supplementary Table 1). Alongside previous dates from the islet of Eilean Dòmhnuill, North Uist[18], these dates demonstrate that a tradition of islet construction began not long after the time when Neolithic practices, including the use of domesticated crops, animals and pottery and the construction of monuments, first appeared in the Outer Hebrides at around 3800–3700 cal BCE[19,20]. Survey and excavation work have revealed that some of these Neolithic crannogs appear too small for anything other than occasional visits, making permanent occupation very unlikely. Their use for lacustrine rituals – potentially even involving mortuary-related activities – has therefore been proposed[10,13]. The argument for a ceremonial function is further supported by the recovery of large numbers of well-made and extensively decorated pots – some near-complete – from the lake beds around them (Fig. 1). Whilst it is now proven that these sites date to the Neolithic period, the activities that took place on or around them remain uncertain. However, a previous study of a small number of ceramic sherds from the islet of Eilean Dòmhnuill on North Uist[3] had already indicated the potential of absorbed lipids to yield new information. The present analysis of preserved lipid biomarkers from vessels excavated from the more recently discovered Neolithic crannogs therefore offers not only an exceptional opportunity to explore human activities connected with these intriguing sites but also to gain wider insights into the presence of cereals and the nature of their use in the Outer Hebrides during the Neolithic.

The vessels analysed here form an exceptional group, associated with potentially ceremonial contexts and comprising many near-complete vessels. The range of vessel forms include traditional Hebridean ridged and non-ridged baggy jars as well as 'Unstan' type bowls and shouldered bowls (Fig. 1). The Hebridean jars form part of a distinctive Hebridean Neolithic pottery tradition, whilst the 'Unstan' type bowls, found in significant numbers at Hebridean islets, are also found at sites in Orkney, where they are frequently associated with tombs. They have also been found on the northern mainland of Scotland[21].

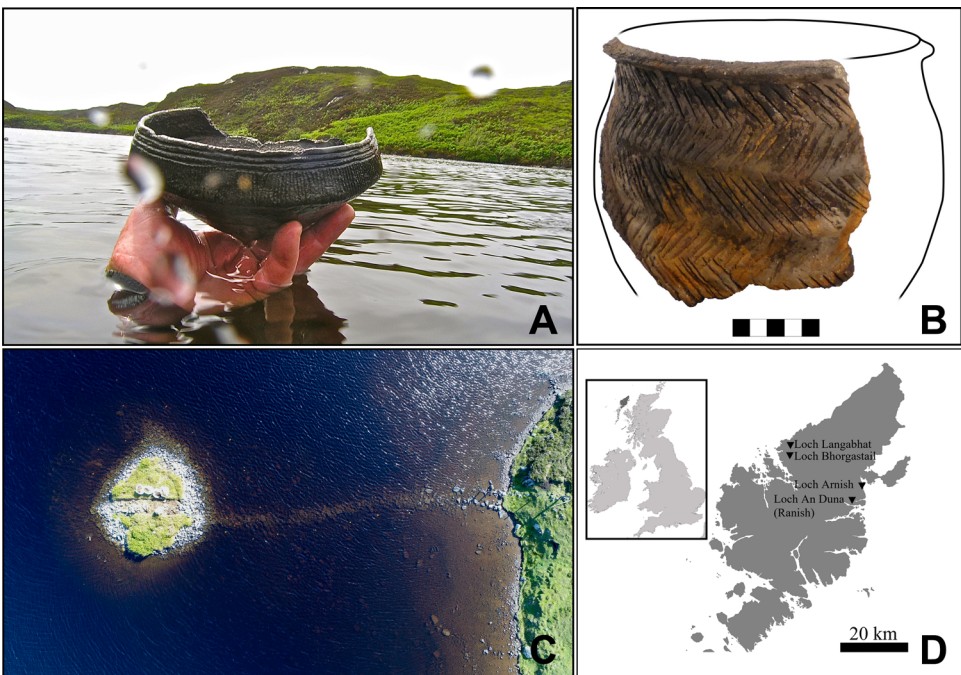

**Fig. 1 | Pottery and sites investigated in this study.** Photographs of an 'Unstan' type bowl recovered from the lake bed at Loch Arnish (**A**), large ceramic sherd and reconstruction of a ridged baggy jar from Loch Langabhat (**B**), aerial view of the islet of Loch Bhorgastail and its associated causeway (**C**) and map of the Isle of Lewis in the Outer Hebrides showing the locations of the four Neolithic crannogs investigated in this study (**D**). The image in Panel A was kindly provided by Chris Murray. Panel **C** was reprinted with permission from University Cambridge Press[10] and the map in Panel **D** was created using Ocean Data View (5.2.1, Reiner Schlitzer, 2020, https://odv.awi.de).

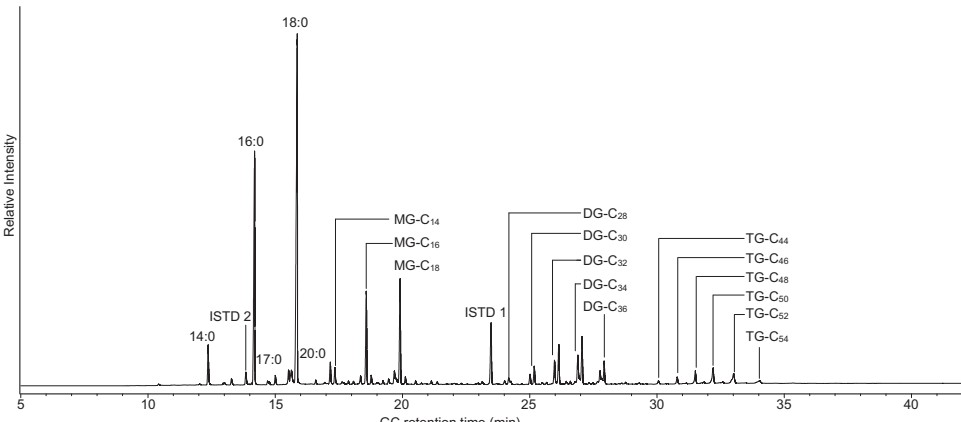

**Fig. 2 | GC-FID chromatogram of the lipid extract from sample LAN16-8a after trimethylsilylation.** Peaks labelled as 14:0–20:0 are saturated fatty acids with 14–20 carbon atoms, while MG, DG and TG denote mono-, di- and triacylglycerols with a total of 14–54 ($C_x$) carbon atoms in the fatty acid chains. ISTD 1 and ISTD 2 denote the internal standards *n*-tetratriacontane and methyl heptadecanoate, respectively.

The high numbers of 'Unstan' type bowls at these possible 'gathering' or ceremonial sites in the Outer Hebrides may therefore point to some sort of ceramic 'lingua franca' whose meanings were understood beyond the Outer Hebrides themselves.

Through the development and application of highly sensitive biomolecular techniques, it is possible to retrieve and characterise chemical traces resulting from the use of ancient pots in the past. The fundamental principle of the analytical approach used in this study, commonly referred to as organic residue analysis (ORA), is that lipids liberated from food cooked or stored in pottery vessels during use, can migrate into the ceramic matrix of unglazed pottery vessels and subsequently can be preserved there[22]. Through detailed molecular and isotopic profiling, the resources contained in the vessel can be deduced and, in this way, past foodways reconstructed[23]. However, plants in general have previously been difficult to detect with this approach, and cereals in particular were invisible. In 2017, Colonese et al. were able to detect cereal-specific lipids extracted from the amorphous, in-situ preserved residues of a Bronze Age wooden container from Switzerland[24]. We have recently demonstrated that these compounds can be absorbed and preserved in the matrix of ceramic vessels and it is also possible, using highly sensitive and selective methods of gas chromatography coupled to high resolution mass spectrometry, to confidently detect and identify traces of these in pottery samples alongside animal-derived lipids[9]. Since the vast majority of the pottery recovered around the Neolithic crannogs in the Outer Hebrides had been deposited directly into the water, this was anticipated to provide ideal anoxic conditions for the preservation of the cereal biomarkers.

Our investigation here focused on the analysis of preserved lipids from vessels recovered from four Neolithic Hebridean crannogs and targeted three key questions: (I) Is it possible to detect diagnostic lipid markers surviving from the use of Neolithic cereals and, if so, can this provide direct evidence for the preparation of this key staple for consumption? (II) Can the organic residues reveal broader patterns in culinary traditions, which might relate to the ways in which different resources were being prepared and consumed? And (III), can this shed light on the activities taking place at these enigmatic sites and their relation to Neolithic practices more generally in the Outer Hebrides and beyond?

## Results

### Lipid recovery and composition of lipid residues
A total of 59 Neolithic sherds were selected from Loch an Duna, Ranish (*n* = 11), Loch Arnish (*n* = 17), Loch Bhorgastail (*n* = 12) and Loch Langabhat (*n* = 19). Lipids were successfully extracted from sherds at all four sites (Supplementary Table 2). Across all four sites the average rate of samples with lipid contents in excess of 5 μg/g (determined by GC-FID) was about 58%. Regardless of the results from GC-FID screenings, all samples were subsequently analysed by GC-QToF MS for cereal biomarkers.

In most lipid residues with detectable quantities of lipids saturated $C_{14}$, $C_{16}$ and $C_{18}$ fatty acids dominated, which is typical for degraded animal fats. In 27 lipid extracts we also detected intact triacyglycerols, frequently alongside their hydrolysis products, i.e. monoacylglycerols and diacylglycerols (Fig. 2, Supplementary Table 2). This indicated very good preservation of lipids and predominantly anoxic conditions at the site due to the waterlogged environment from which the pottery was recovered. In seven samples midchain ketones ($C_{29}$–$C_{37}$) were detected, which form in these distinctive distributions through a ceramic-catalysed condensation of two fatty acids[25,26]. Since this reaction only occurs under substantial heat (>250 °C), their presence can be used to infer significant heating of the lipids in the vessels in question, which could be related to cooking practices (e.g. roasting) or to post-firing treatment of the pots[26,27].

### Identification of animal lipid sources using δ13C analysis of fatty acids
To achieve further characterisation of the lipid residues and identify the major animal lipid source the $δ^{13}C$ values of the $C_{16:0}$ and $C_{18:0}$ fatty acids were determined by GC-C-IRMS in a total of 29 samples (including one visible residue). It is known that environmental effects can shift the absolute $δ^{13}C$ values of these fatty acids relative to the reference values, which were determined using animals raised on a purely $C_3$ terrestrial diet from mainland southern Britain[28]. Such a shift towards less depleted isotopic compositions was also observable here, particularly in samples from Loch Bhorgastail and Loch Langabhat (Fig. 3C, D). To account for this shift and to aid interpretation the $Δ^{13}C$ ($δ^{13}C_{18:0}$ - $δ^{13}C_{16:0}$) values were plotted against the $δ^{13}C_{16:0}$ values[28], and both plots together are used for data interpretation (Fig. 3, Supplementary Fig. 1). Consistent with previous reports from Neolithic sites in Scotland and elsewhere in Britain was a dominance of lipid signatures consistent with ruminant carcass and dairy fats[3,28–30]. In fact, more than half of the samples investigated by GC-C-IRMS showed $δ^{13}C$ values consistent with the reference values for pure dairy fats (Fig. 3, Supplementary Table 2). The identification of dairy residues could also, in many cases, be supported by triacylglycerol patterns, which showed a wide distribution from $C_{40}$ to $C_{54}$ which is typical for degraded dairy fats[31]. Isotopic analysis provided no evidence for non-ruminant (e.g. porcine) lipids or an input of aquatic lipids, but a low admixture could

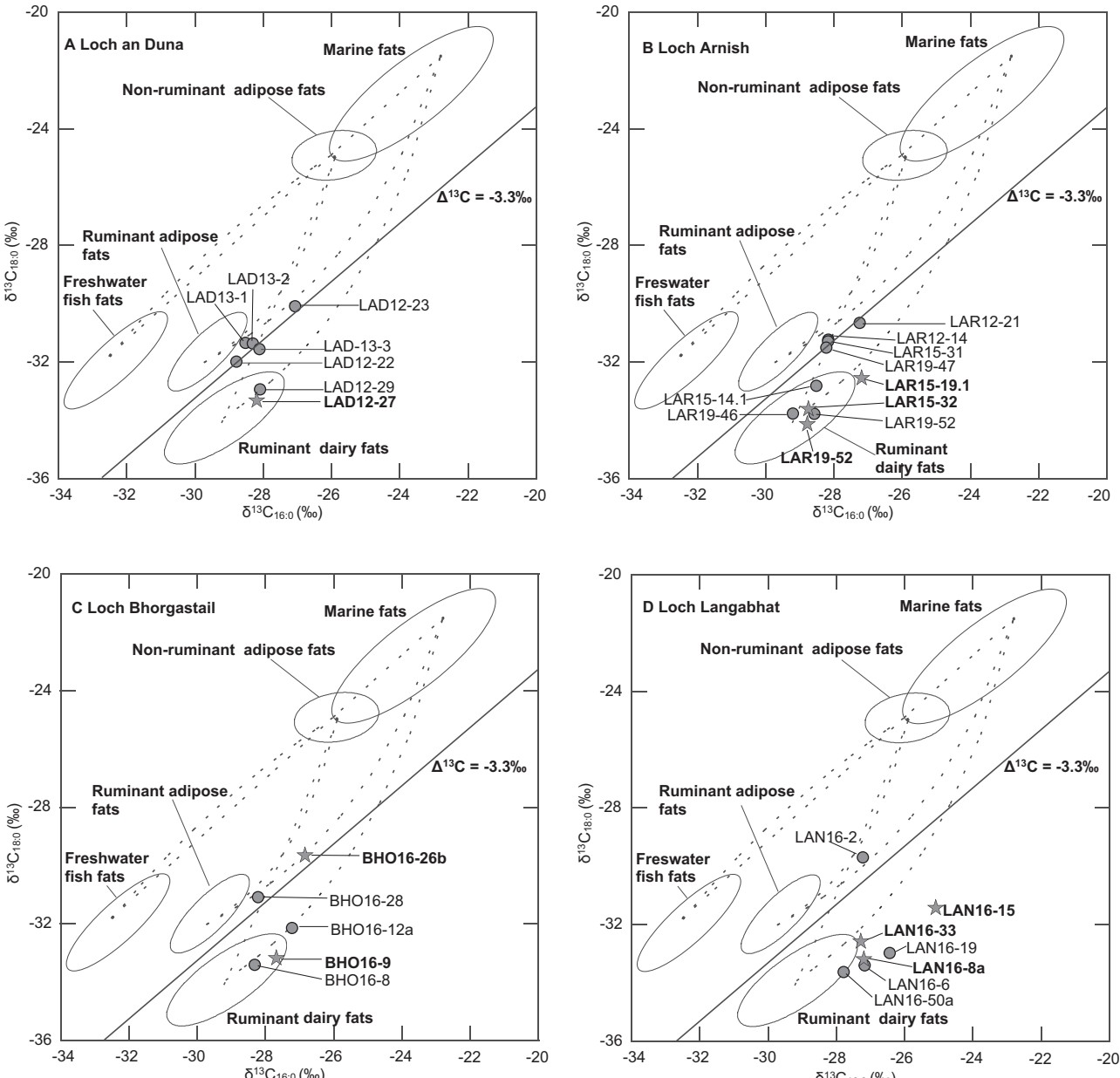

**Fig. 3 | Scatter plots of δ¹³C values of C₁₆:₀ and C₁₈:₀ fatty acids determined by GC-C-IRMS.** The plots show the δ¹³C values of C₁₆:₀ and C₁₈:₀ fatty acids prepared from total lipid extracts of the pottery from Loch an Duna (**A**), Loch Arnish (**B**), Loch Bhorgastail (**C**) and Loch Langabhat (**D**). All δ¹³C values obtained for modern reference terrestrial animal fats were from animals raised on pure C₃ diet[28] and have been adjusted for the post-Industrial Revolution effects of fossil fuel burning, by the addition of 1.2‰[47]. Reference values for aquatic fats are from marine species from UK waters, Atlantic and North Sea (marine) and a UK lake and Kazakh river (freshwater fish)[48]. The diagonal line separates the reference range of ruminant adipose and dairy fats. Lines connecting the ellipses represent theoretical δ¹³C values obtained through the mixing of these fats and are for orientation only. Extracts with cereal biomarkers are displayed as stars with bold labels. Source data are provided as a Source Data file.

have been masked by the isotopic signal of the more abundant ruminant lipids.

No animal bones have been recovered from any of the sites, with acidity levels here (as at many other terrestrial excavations in the region) not being conducive to preservation of osseous material. Therefore, lipid residues provide unique information on the use of these vessels and the exploitation of animal products that would otherwise not be available.

A slight difference could be observed between residues extracted from pots from the west of the island (Loch Langabhat and Loch Bhorgastail) on one hand and from the east of the island (Loch an Duna

and Loch Arnish) on the other. In lipid extracts from Lochs Langabhat and Bhorgastail, the isotopic values formed two separate clusters, with three residues showing Δ¹³C values consistent with ruminant carcass fats (between −2.5 and −2.2 ‰), while all others exhibited Δ¹³C values of −6.3 to −4.6‰, which is in the reference range of pure dairy fat (Fig. 3C, D, Supplementary Fig. 1). In contrast to this, at Loch an Duna and Loch Arnish no residues fell within the reference range for ruminant carcass fats, but several were more likely to reflect a mixture of ruminant carcass and dairy fats. The range of Δ¹³C was from −5.4 to −2.8‰ and therefore significantly narrower than that seen at the other two sites (Supplementary Table 2, Fig. 3A, B).

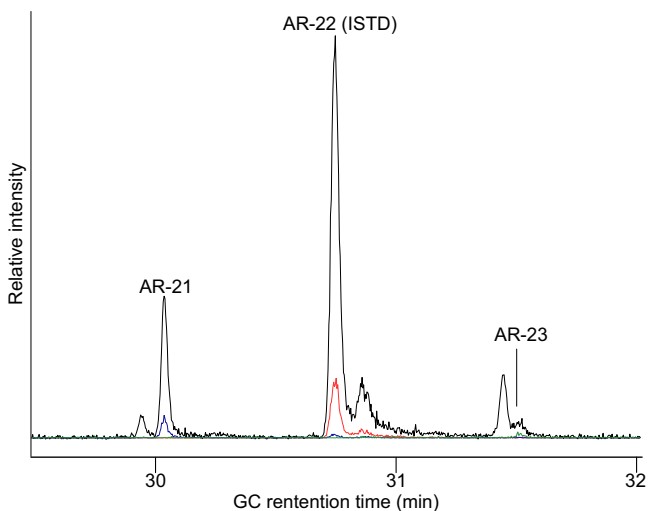

**Fig. 4 | Partial GC-QToF MS Extracted Ion Chromatograms showing the elution of alkylresorcinols in extract LAN16-33.** Drawn in black is the Extracted Ion Chromatogram of $m/z$ 268.1315, the base peak in spectra of trimethylsilylated alkylresorcinols (AR). Drawn in blue, red, and green are Extracted Ion Chromatograms of $m/z$ 548.4445, $m/z$ 562.4602 and $m/z$ 576.4833, which are the molecular ions of AR-21, the internal standard AR-22 (ISTD, added at level of 50 ng/g to samples) and AR-23, respectively. Both ions in combination with the retention time allow the unambiguous identification of the alkylresorcinol homologues.

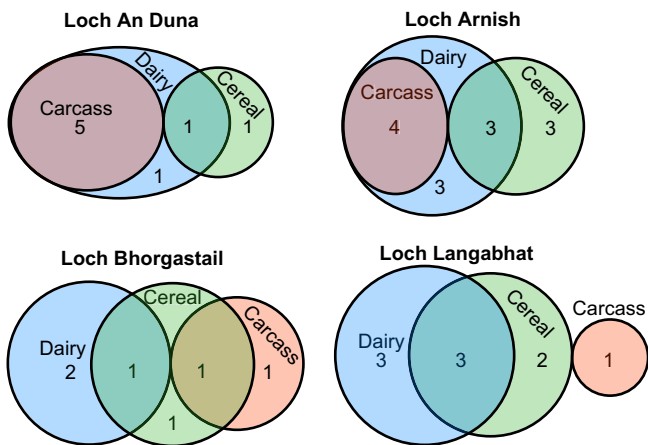

**Fig. 5 | Venn Diagrams showing the distribution and mixing of ruminant carcass (red) and dairy fat (blue) as well as cereal lipids (green) in lipid extracts from Loch an Duna, Loch Arnish, Loch Bhorgastail and Loch Langabhat.** Sizes of the ellipses correspond with relative proportions of the commodity for each individual site and numbers denote the absolute number of samples with lipid signatures characteristic for this respective commodity or combinations of these at each site. This illustrates the absence of mixing of carcass and dairy products in pots from Lochs Bhorgastail and Langabhat, unlike their frequent co-occurrence in samples from Loch an Duna and Loch Arnish, and the overall very rare co-occurrence of cereal biomarkers with carcass (meat) products. Source data are provided as a Source Data file.

## Evidence for processing of cereals and other plants

Cereal lipid biomarkers, i.e. alkylresorcinols, alongside plant sterols and stanols, were detected in a total of 16 samples (including two visible residues) from the four sites, which suggests a widespread use of cereals (Supplementary Table 2). Alkylresorcinols are a group of compounds characteristic for cereals consisting of a common resorcinol backbone and a variable $C_{15}$ to $C_{27}$ alkyl chain, which is basis for shorthand naming of the individual homologues (the shorthand name for the homologue with a $C_{21}$ alkyl chain, i.e. 5-*n*-heneicosylresorcinol, is AR-21). While in samples from Loch an Duna (two extracts), Loch Arnish (six extracts) and two out of three extracts from Loch Bhorgastail only the homologue AR-21 was detected, samples from Loch Langabhat (five extracts) and one extract from Loch Bhorgastail also featured the homologue AR-23 and (one extract) AR-19 (Fig. 4). In all cases bar one from Loch Bhorgastail (BHO-16-26a) AR-21 was the dominant homologue. Levels found in extracts from Loch Langabhat were generally higher than at the other three sites but nonetheless were still <50 ng/g ceramic. Considering the low quantities of these compounds distinguishing genuine food-related and environmental sources is challenging, but the specific association of the cereal biomarkers with certain vessel shapes and contents, as well as mismatch with the patterns of lipid biomarkers in environmental samples, makes an environmental source very unlikely (see Supplementary Discussion).

In modern cereal samples the alkylresorcinol pattern can be used for discrimination of different cereal species (e.g. different species of *Triticum* sp., such as bread wheat: *Triticum aestivum* L. ssp. *aestivum* from einkorn wheat: *Triticum monococcum* L. subsp. *monococcum*). However, this has to be approached with caution for archaeological cereal residues since patterns can change during cooking and possibly post-depositionally[9,23,32,33] (See Supplementary Discussion). Still, considering the dominance of AR-21 (which is characteristic for wheat species) and the total absence of AR-25 (the dominant AR homologue in *Hordeum* sp.: barley[34]), and previous experiments on the uptake and preservation of ARs, the pattern present in the residues indicates the processing of some *Triticum* species and not barley, which is the only other cereal relevant to our study region[4]. Only very limited numbers of cereal macrofossils (none from the Neolithic) or other

archaeobotanical remains have been recovered from Loch Langabhat and Loch Bhorgastail so far (none for the other two sites, which have not been excavated), but extensive analysis of archaeobotanical data from across Scotland reveals that the available *Triticum* sp. crops in this period include emmer wheat (*T. turgidum* L. ssp. *dicoccum* (Schrank) Thell.) and free-threshing wheat (*T. aestivum ssp. aestivum/ T. turgidum L. ssp. durum/T. turgidum L. ssp. turgidum*), with emmer the dominant wheat crop[4]. A single find of einkorn wheat has been uncovered from a Neolithic context in Scotland at Varme Dale on Orkney[35], but this has not been directly radiocarbon dated and the identification is equivocal. Current evidence therefore suggests that einkorn was largely absent from Scotland at this time, as elsewhere in Britain[4,6,8]. Hence, it is most likely that the *Triticum* sp. cereal lipid biomarkers derive from emmer or free-threshing wheat.

Of the 16 extracts that provided evidence for cereal processing eight samples also showed clear signatures of dairy fat, and one sample had both evidence for cereal processing and an isotopic signature consistent with ruminant carcass fats (Supplementary Table 2), indicating possible mixing of those commodities (Fig. 5). Five samples (including sample LAN16-32, which showed the highest abundance of ARs), were found in initial GC-FID screening to have total lipid contents lower than 5 μg/g, i.e. the commonly applied threshold of interpretable lipid contents. The remaining samples had total lipid contents slightly higher than 5 μg/g but the major lipid source could not be determined due to the low concentration of fatty acids deriving from animal fats. Furthermore, it should be noted that both visible residues yielded detectable quantities of alkylresorcinols, even though they were not detected in the absorbed residues from the respective ceramic samples, despite relatively similar $\delta^{13}C$ values between both residues in the case of sherd LAR19-52, indicating similar compositions. This demonstrates the very limited absorption of these compounds into the ceramic matrix, which has been observed before[9]. It could be helpful, in future research, also to screen visible residues for cereal biomarkers, while bearing in mind the susceptibility of surface residues for contamination.

Traces of plant sterols were detected without alkylresorcinols in numerous lipid extracts (Supplementary Table 2). This could either be due to a preferred loss of alkylresorcinols over plant sterols in some

samples or it could indicate the processing of plant foods other than cereals, as phytosterols occur in essentially all plants. Widespread evidence for the collection of edible wild plants, such as hazelnuts for instance, has been recovered from Neolithic sites across Scotland[4] and it is possible that such wild plant processing is represented here. Sterols are very seldom reported in archaeological samples and, in the case of cholesterol, are usually suspected to be modern contamination from fingerprints[36]. The widespread detection of these plant sterols was also surprising in light of our recent experiments showing the catalytic properties of ceramics promoting sterol degradation under heat[37]. In fact, both cholesterol, which was also detected in a number of samples, and phytosterols were detected alongside more abundant hydrogenation as well as oxidation products (Supplementary Fig. 2).

## Discussion

Our results represent the first direct evidence for the cooking of cereals in Neolithic pots from the 4th millennium BCE, based on specific molecular lipid markers. This demonstrates how our approach based on gas chromatography and high resolution mass spectrometry can be used for comprehensive screening of lipid extracts and, especially in combination with prior enrichment, for the highly sensitive and selective analysis of target compounds. Importantly, this approach provides the superb sensitivity to target biomarkers present at levels of only few ppb in the ceramics without, in contrast to Selected Ion Monitoring or Multiple Reaction Monitoring, losing information on the other compounds present in the sample. In addition to now enabling the detection of this important commodity, the application of this approach more widely in archaeological science offers vast opportunities in terms of wealth of information that can be gained from each individual sample and even allows completely new angles of interpretation to be considered[38].

With this research we have also demonstrated that cereal-specific lipid biomarkers can be preserved under favourable conditions for long enough to allow them to be used to investigate culinary practices and associated human behaviour as far back as the Neolithic. Only from Anatolia has earlier biomolecular evidence for cereal processing been recovered from pots, and this has relied on the presence of cereal-specific proteins from calcified deposits[39], which are relatively uncommonly recovered archaeologically. Such biomolecular evidence is especially important in sites or regions where other, more traditional archaeological indicators of plant cultivation (such as charred plant macrofossils) have not been preserved and in contexts where opportunities for past cereal macrofossil deposition are limited. Relatively few Early Neolithic sites have been sampled for archaeobotanical remains in the Outer Hebrides so far. Excavated deposits at Loch Langabhat were sampled for charred plant remains but the single cereal grain recovered was of indeterminate species and was associated with Middle Bronze Age occupation deposits[15]. Our biomarker-based analyses indicated the processing of wheat (and not barley) at all four sites. This is in stark contrast to other data concerning cereal grain assemblages recovered across the broader region of Scotland during the Neolithic, which (with but a couple of Early Neolithic exceptions) show a notable dominance of barley over wheat, especially within this Atlantic region[4]. While both emmer wheat and free-threshing wheat grains have been identified in Neolithic samples from the Outer Hebrides, the small quantities so far recovered have suggested that wheat was a weed contaminant of the barley crop rather than a deliberately cultivated crop. Our results are therefore both surprising and significant. It has to be considered that the AR content in barley is about tenfold lower than in wheat, meaning that the quantities transferred to the ceramic matrix might be too low for detection and may therefore not be detected at all (if used alone), or not detectable next to the detected wheat signature (if used together)[32]. Nonetheless, our findings clearly demonstrate that wheat was consumed at an early stage at the onset of farming within this region - as well as barley, as indicated by the

wider archaeobotanical data from across the region. Our data are therefore in apparent contrast to the archaeobotanical record, which has suggested that very little wheat was consumed in Atlantic Scotland throughout the Neolithic[4]. As mentioned above, given the tenfold higher abundance of ARs in wheat compared with barley, these findings do not preclude the processing of barley alongside wheat in these pots. However, as the current role of wheat versus barley is currently unknown in the Neolithic Outer Hebrides in comparison with wider Atlantic Scotland, our approach appears to have rendered substances visible that have in the past not been identified using traditional modes of analysis. In the future, our methods may therefore allow cereal consumption to be identified at sites where little crop processing/fire-setting had taken place, and/or where specific processing and cooking practices resulted in little chance of accidental grain carbonisation and preservation, such as ceremonial or funerary contexts. For instance, it could be that the wheat and barley were prepared for consumption in different ways in this region, resulting in different traces of past consumption in the archaeological record. Wheat may have been boiled in soups or porridges for example, resulting in little accidental grain carbonisation but representation in lipid residues, whereas barley may have been dry roasted on hot stones for consumption, a practice which frequently results in considerable grain carbonisation[40]. Therefore, using this combined approach, with complementary data from both archaeobotany and AR, significantly strengthens our interpretations about early food consumption practices.

Having demonstrated that evidence for a 'missing' part of the Neolithic package of domesticated plants and animals can survive in pottery organic residues, we now consider more holistically the ways in which different resources may have been prepared. Firstly, and most notably, we have observed that cereals were very rarely detected in the same pots that also had evidence of animal carcass lipids. Despite finding 16 occurrences of cereal biomarkers, these were found either in pots otherwise containing only dairy fats ($n = 8$) or in pots without evidence for accompanying animal products at all ($n = 7$), and only in one instance were cereal biomarkers recovered from a pot containing ruminant carcass fat (Fig. 5). Since our previous experiments have shown that cooking cereals alongside meat would actually result in the most effective transfer of alkylresorcinols into the pot matrix[9], this pattern is unlikely to result from taphonomic factors. Markers of more generic plant origin (not specifically attributable to cereals) were, however, found widely across all vessel types analysed. It therefore appears that cereals were likely either boiled in jars used more-or-less exclusively for this purpose, or cooked in jars used for milk; it is quite plausible that both elements were conducted together (i.e. as a type of porridge/milk gruel), although sequential uses of these pots for dairy products and boiling of cereals is also possible. Noteworthy, cereal lipids also contain the fatty acid $C_{16:0}$ and a relatively low proportion of $C_{18:0}$. If a substantial proportion of cereals were prepared in pottery alongside low quantities of meat, this would draw the the $C_{16:0}$ fatty acid to exhibit a more depleted $\delta^{13}C$ value[23]. Given the relatively low quantity of $C_{18:0}$ fatty acid in cereal lipids, in this scenario the $C_{18:0}$ fatty acid would rather retain a predominantly animal-derived signature, thus reducing the $\Delta^{13}C$ value and potentially masking a dairy fat contribution. Given that the $\delta^{13}C$ values of these residues are not notably depleted, and dairy fats are widely identified, we do not think this scenario plausible in this instance.

Secondly there is a clear correlation between the rim diameter of vessels (where measurable) and vessel contents (Fig. 6). Those with smaller rim diameters of <25 cm (including bowls and smaller jars) were used for dairy products (alongside plants and cereals) whilst jars with larger rim diameters (>30 cm) were used for meat, probably with some contribution of dairy products as well, in addition to plants (without evidence for cereals). Importantly, however, there is not a straightforward correlation with pot volume; whilst the bowls are relatively small in volume (<1 L), jars of the same diameter are

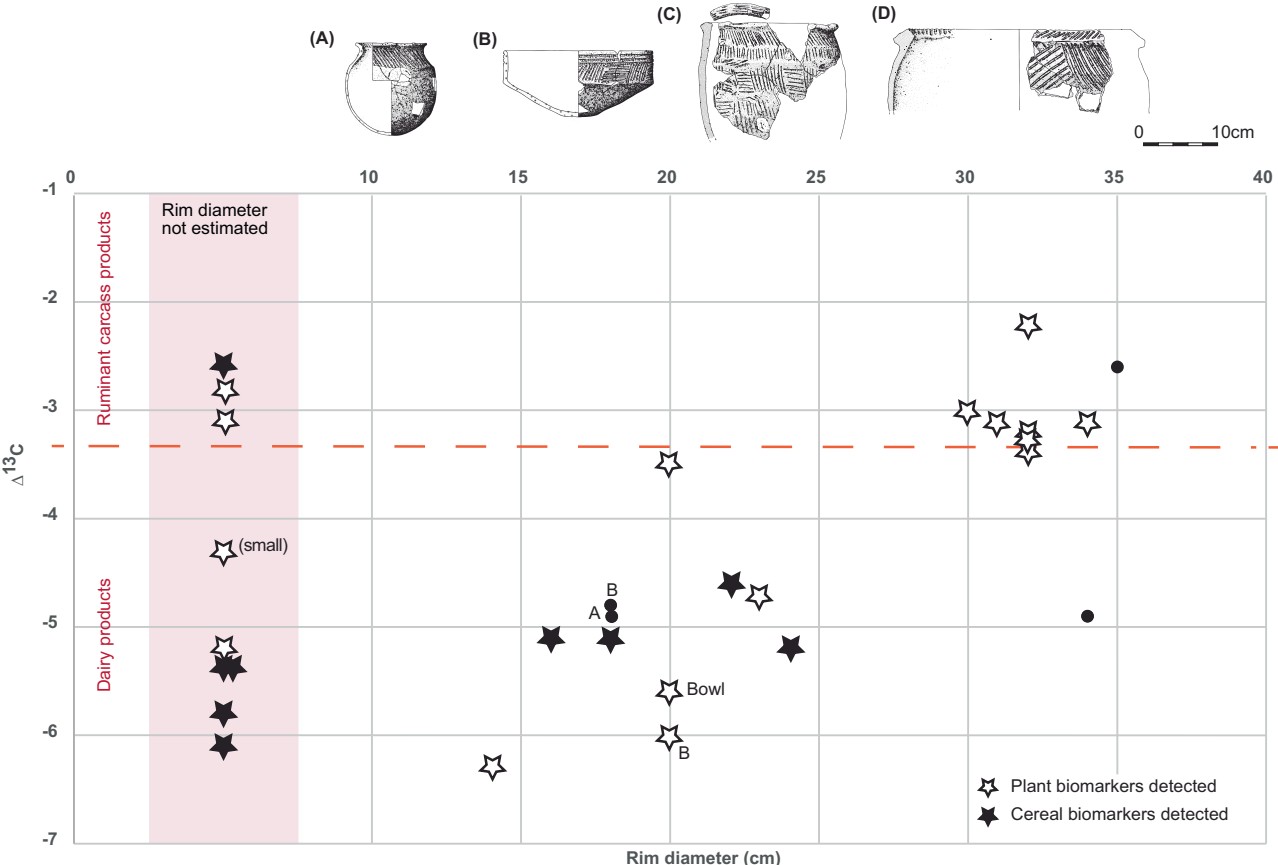

**Fig. 6 | Plot showing the $\Delta^{13}C$ ($\delta^{13}C_{18:0}$ – $\delta^{13}C_{16:0}$) value of the fatty acids from degraded animal fats plotted against rim diameters of Neolithic pots, where possible to estimate.** Data points shown as stars denote the additional observation of plant-derived markers in the organic residue, whilst a filled star denotes the presence of diagnostic cereal-derived markers. The dashed line shows the approximate boundary between predominantly dairy and predominantly carcass fats of $\Delta^{13}C = -3.3$ ‰. Vessel forms show examples of various categories including: **A** – necked bowl, **B** – Unstan-type bowl, **C** – ridged baggy jar and **D** – larger ridged baggy jar, from the nearby islet of Eilean Domnhuill. All data points on the graph are jars, unless otherwise labelled. Source data are provided as a Source Data file.

significantly deeper and therefore have a considerably larger (e.g. at least 3x) volume. It is therefore possible that the vessel form (but not size per se) dictated its use – i.e. wider-mouthed vessels were used for meaty stews and mixed cooking, whilst both shallow bowls and jars with relatively narrow mouths were used for milk and cereals. Interestingly, this can be compared and contrasted with patterns seen in Middle Neolithic pottery from eastern France where dairy products were associated either with small pots (micro pots or pots with <20 cm rim diameter and volumes of <1 L) or large open-mouthed vessels with a much larger volume distribution including open cups and goblets[41]. Finally, it can be observed that the relatively small 'Unstan' type bowls appear to be associated with dairy products and not meat-based dishes. The number of data points is low; however, at this stage it would appear unlikely that these bowls were used as serving bowls for the meaty stews being cooked in the larger jars, but rather that they were reserved for a separate purpose. Moreover, wider usage patterns relating to the pottery deposited at crannogs from two different areas of Lewis are emerging, with evidence that pots deposited at Loch Bhorgastail and Loch Langabhat (in the west) were routinely reserved for meat *or* dairy products, compared with more regular mixing of both from pots recovered from Loch an Duna and Loch Arnish (in the east) (Figs. 1 and 5).

Finally, we consider these findings in light of their implications for the activities taking place at these crannog sites and in the Outer Hebrides more broadly during the Neolithic. Our results may imply that traditional archaeobotanical sampling has yet to identify the full diversity of crops cultivated during the earliest phase of farming in this region, a realisation that has significant implications for all site types

across the whole of Scotland and beyond. Furthermore, the presence of wheat specifically at Neolithic islet sites raises interesting additional questions. If the wheat was cooked in the pots at the crannogs, this finding could be viewed as adding weight to the argument that these were 'special' sites, with unusual foodstuffs being preferentially chosen for consumption there before the vessels which contained them were deposited into the water. However, given the lack of association between specific cereal crops and clear-cut ceremonial and/or ritual contexts elsewhere in Scotland and the small number of sites in the Outer Hebrides with sampled archaeobotanical remains, a more likely explanation is perhaps that the pots were brought to the crannogs having already been in use at domestic settlements in the vicinity. Our results probably therefore reflect very localised patterns of dietary behaviour – amongst those people using these crannog sites – that seemingly contrast with plant macrofossil evidence from other parts of the Outer Hebrides and Atlantic Scotland more broadly[4]. Due to the preservation conditions required for survival, along with specific protocols for the concentration and detection of cereal-specific biomarkers, such evidence for cereal processing has unsurprisingly not been detected in the relatively low number of sherds that have been analysed from other Neolithic sites in the Outer Hebrides. However, these earlier results, which indicate widespread presence of dairy products from both another islet (Eilean Dòmhnuill) and a sea stack (Dunasbroc, possibly a cliff top site during the Neolithic)[42], are consistent with a general prevalence of milk-derived products in vessels from this region[3].

As discussed at the start of this article, crannogs in Scotland – and especially these newly-discovered Neolithic examples – remain an

enigmatic type of site. Many Neolithic examples appear too small to have been settlements and have thus far failed to produce clear evidence for buildings dating to the Neolithic. Other Neolithic occupation sites have been identified within the wider landscape of Lewis[19] but these are, without exception, small-scale, ephemeral sites whose nature is not at present well understood. The relationship between these occupation sites and the Neolithic islets is in need of further detailed investigation. It is likely, however, that the people who used the crannogs lived elsewhere in the local region and may even have regularly moved from site to site. The 'liminal' location of these crannogs, out on the water and away from everyday life, their potentially 'monumental' forms and the deposition of large quantities of pottery in the water around them add weight to the suggestion that these islets could have been associated with specialist, potentially ceremonial, behaviours, with feasts and other communal activities being carried out there[10,13]. New information gained from our study of the ceramic-preserved biomarkers provides compelling new insights – not only about the culinary traditions and behaviours of those people using these sites and depositing pots into the lochs around them, but also about the lifeways of these early farming communities in this region at the north-western edge of Europe.

## Methods

### Pottery sherds
A total of 59 sherds from the four sites, Loch an Duna (Ranish, $n = 11$), Loch Arnish ($n = 17$), Loch Bhorgastail ($n = 12$) and Loch Langabhat ($n = 19$) were analysed for absorbed lipid residues (Supplementary Table 2). Permission to carry out destructive sampling was granted by the Scottish Archaeological Finds Allocation Panel and Queen's and Lord Treasurer's Remembrancer as part of the Treasure Trove process. In addition, two visible residues from Loch Arnish were analysed. This material was characteristic of the Early/Middle Neolithic 'Hebridean' style (ca. 3700–3000 BCE), consisting primarily of baggy and ridged jars with occasional 'Unstan' type bowls as well[43]. All of this material is described in detail within recent interim reports[15,17]. In each case, the ceramic material appears to have been deposited directly into the loch, adjacent to the crannog, in prehistory. The soft silts and relatively low-energy hydrological environment have led to excellent preservation of the pottery, and apparently also the organic residues within. Following recovery of this material from the loch bed, the pottery was stored according to standard archaeological practice in plastic bags within cardboard storage boxes. Nomenclature for the cereals follows Zohary et al.[44].

### Chemicals
Chloroform, dichloromethane (DCM), $n$-hexane, methanol and ethyl acetate (all HPLC grade) were from Rathburn (Walkerburn, UK). $n$-Tetratriacontane ($C_{34}$, >98%), pyridine (>99%), the silylating agent consisting of $N,O$-bis(trifluoroacetamide)/trimethylchlorosilane (BSTFA/TMCS) 99:1 (v/v), borontrifluoride reagent (14 %wt $BF_3$ in methanol), silica for chromatography and methyl heptadecanoate (17:0-ME, >99%) were from Sigma-Aldrich (Munich, Germany). An authentic alkylresorcinol standard (5-$n$-docosylresorcinol, AR-22, >99%) was supplied by ReseaChem (Burgdorf, Switzerland). Aminopropyl-bonded bulk sorbent for solid phase extraction (Isolute, 50 μm particle size) was purchased from Biotage (Uppsala, Sweden).

### Sample preparation and lipid extraction
Ceramic samples were cleaned using a modelling drill and powdered using a DCM-washed mortar and pestle. About 2 g of ceramic powder were spiked with 40 μg of $C_{34}$ and 100 ng of AR-22 and extracted using $2 \times 10$ mL of chloroform/methanol 2:1 (v/v) under sonication. The solvent was removed under a gentle stream of nitrogen at 40 °C and the residue was redissolved in 2 mL of chloroform/methanol 2:1 (v/v). An aliquot of 500 μL of this total lipid extract (TLE) was applied to a glass

column (1 cm i.d.) filled with ca. 0.5 g of silica (preconditioned with 5 mL chloroform/methanol 2:1 (v/v)). Lipids were eluted using 5 mL chloroform/methanol 2:1 (v/v), the solvent was transferred to a vial and removed under a gentle stream of nitrogen. Pyridine (25 μL) and the silylating agent (50 μL) were added, the vial was sealed and heated at 70 °C for 1 h. Afterwards, the silylating agent was removed under a gentle stream of nitrogen. The residue was redissolved in $n$-hexane (0.5 mL) and after addition of the third internal standard (17:0-ME) was analysed by GC-FID and subsequently GC-QToF MS.

Fatty acid methyl esters for GC-C-IRMS analysis were prepared from total lipid extracts using a procedure based on $BF_3$/MeOH[45]. Aliquots of the TLEs were transferred into glass culture tubes and the solvent was removed using a gentle stream of nitrogen. For saponification, the residue was re-dissolved in 2 mL of 0.5 M potassium hydroxide in methanol and after thorough mixing, the solution was heated at 70 °C for 1 h. After cooling to room temperature, 1 mL of ultrapure water was added, the pH was adjusted to pH 3 using 1 M HCl solution and the fatty acids were extracted using $3 \times 3$ mL chloroform. The chloroform phases were combined in a new culture tube and the solvent was removed using a gentle stream of nitrogen. To the residue 100 μL of boron trifluoride (14%wt in methanol) were added, and the tube was heated at 70 °C for 1 h. Afterwards, 2 mL of water was added and the fatty acid methyl esters were extracted using $3 \times 2$ mL chloroform, the solvent was removed and the residue was redissolved in 1 mL of $n$-hexane and screened by GC-FID to adjust concentrations for GC-C-IRMS analysis.

### Enrichment of cereal biomarkers by solid phase extraction
For samples with a high abundance of free fatty acids or high molecular weight lipids (mainly triacylglycerols) that would not be amenable to regular (i.e. not high temperature) GC-MS, an extraction of cereal biomarkers from the total lipid extract was performed. For this purpose, 500 μL of the TLE was transferred to a vial, the solvent was removed under a gentle stream of nitrogen, and the residue was redissolved in ca. 500 μL $n$-hexane/ethyl acetate 96:4 (v/v). A glass column (1 cm i.d.) was filled with 0.5 g of aminopropyl sorbent and equilibrated with $2 \times 5$ mL $n$-hexane/ethyl acetate 96:4 (v/v) before the sample was added. The first fraction was eluted with $2 \times 5$ mL $n$-hexane/ethyl acetate 96:4 (v/v) and discarded before cereal biomarkers were eluted using $2 \times 5$ mL ethyl acetate/MeOH 90:10 (v/v) (Supplementary Fig. 3). The solvent was removed using a gentle stream of nitrogen and the residue was trimethylsilylated as described before. After redissolving in $n$-hexane (200 μL) the solution was screened by GC-FID and analysed by GC-QToF MS.

### Analysis of lipids by GC-FID, GC-QToF MS and GC-C-IRMS
Analysis of trimethylsilylated lipid extracts by GC-FID after silica or aminopropyl clean-up as well as fatty acid methyl esters prior to GC-C-IRMS analysis was performed as using a 7820 GC-FID system equipped with a 15 m, 0.32 mm i.d., 0.1 μm film thickness DB-1HT column (Agilent Technologies, Santa Clara, CA/USA)[9]. Sample aliquots (1 μL) were injected using a cool-on-column inlet. After 2 min at 50 °C, the temperature was increased to 350 °C at a rate of 10 °C/min. This final temperature was held for 10 min. Helium was used as carrier gas at a constant flow rate of 4 mL/min while the flow rates of hydrogen, synthetic air and nitrogen for the flame ionisation detector (350 °C) were set to 30, 400 and 27 mL/min, respectively. Data were collected and analysed using Chemstation (version B.03.02, Agilent Technologies, Santa Clara, CA/USA).

Subsequent analysis of lipid extracts by GC-QToF MS was performed using a 7890/7200B GC-QToF MS system equipped with a 50 m × 0.32 mm i.d., 0.17 μm film thickness HP-1 column (Agilent Technologies, Santa Clara, CA/USA) and a multimode inlet operated in splitless mode[9]. After 2 min at 50 °C, the temperature of the oven was increased to 320 °C at a rate of 20 °C/min. This final temperature was

held for 20 min. Helium was used a carrier gas at a constant flow rate of 1.5 mL/min. Mass spectrometry data was collected from $m/z$ 50 – 1050 at a rate of 5 scans/s using the *Extended Dynamic Range mode* with the ion source, quadrupole and transfer line temperatures set to 230, 150 and 320 °C, respectively. Data were collected using MassHunter (version B.07.02.1938, Agilent Technologies, Santa Clara, CA/USA) and analysed using Qualitative Analysis (version B.07.00, Agilent Technologies, Santa Clara, CA/USA) and MZmine 2.5 (mzmine.github.io).

For compound specific $\delta^{13}C$ analysis, fatty acid methyl esters were analysed by GC-C-IRMS using a 7890A GC (Agilent Technologies, Santa Clara, CA/USA), an IsoPrime GC5 combustion interface and an Iso-Prime 100 IRMS (Elementar, Cheadle, UK)[45]. A 50 m × 0.32 mm i.d., 0.17 μm film thickness HP-1 column (Agilent Technologies, Santa Clara, CA/USA) was used for the separation and samples were injected using a SSL injector in splitless mode. The GC oven program was as follows: After 2 min at 40 °C, the temperature was raised to 300 °C at a rate of 10 °C/min, and this final temperature was held for 10 min. A quartz tube with copper oxide pellets at 850 °C was used in the combustion reactor. All samples were measured in duplicate with deviations in $\delta^{13}C$ values <0.3‰ for all reported samples. Data were collected and analysed using Ion Vantage version 1.6.1.0 (IsoPrime).

### Reporting summary

Further information on research design is available in the Nature Research Reporting Summary linked to this article.

## Data availability

All data needed to support the conclusions of the paper are presented in the paper or the Supplementary Material. The raw GC-QToF MS data have been deposited in the Bristol Research Data Repository under following link: https://data.bris.ac.uk/data/dataset/fn4ujbvbe4nr2eji3icdzvp65[46]. Source data for figures are provided with this paper. All ceramic samples analysed in this project are currently stored at the Universities of Reading and Southampton, pending final deposition at the end of the project according to the Scottish Treasure Trove process. For access to the samples please contact D.G. or F.S. Source data are provided with this paper.

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

## Acknowledgements

We are all extremely grateful to Chris Murray and Mark Elliott through whose vision and hard work these sites were identified (and much of the material we have analysed recovered) in the first place. This study was supported by the Natural Environment Research Council (NERC, NE/N011317/1, S.H. and L.J.E.C.) and the Arts and Humanities Research Council (AH/S010157/1, D.G and F.S). The survey and excavation work of D.G. and F.S. on these sites (2015-17) was generously funded by the British Academy, Leverhulme Trust, Honor Frost Foundation and the Society of Antiquaries of London. The authors wish to thank NERC for partial funding of the National Environmental Isotope Facility (NEIF, contract number NE/V003917/1) as well as NERC and the University of Bristol for supporting its GC-MS (2014 Strategic Environmental Science Capital Call award no. CC010) and GC-IRMS capabilities.

## Author contributions

S.H. and L.J.E.C. conceived the project. S.H., C.G and H.L.W. performed experimental work, D.G. and F.S. excavated the sites, A.S., D.G. R.B. and F.S. provided samples and archaeological background. S.H., L.J.E.C., D.G., L.H. and M.C. analysed the data. S.H., L.J.E.C. and D.G. wrote the manuscript. All authors edited and approved the manuscript.

## Competing interests

The authors declare no competing interests.
