## [Peer Review File · Nature Communications]

Reviewers' Comments:

Reviewer #1:

Remarks to the Author:

This is an interesting paper and makes a major advance in the detection, contextualisation and use of cereals in prehistory. Whilst the approach is not entirely novel, previous studies have demonstrated ARs in wetland contexts dating to the Roman period and in a prehistoric container from the alps, this is the first time it has been applied to prehistoric ceramics which are the most abundant artefact in the archaeological record for this type of research. It also addresses an important area of research concerning the establishment of agriculture in the British Isles and should serve as an important reference for future studies. While the sample size is low, the observation of cereals with vessel size and the co-occurrence with dairy fats provides new insight into cereal use during this period.

The methodology is sound and builds upon previous work by the authors which has validated the approach on experimental material. One concern is the low quantity of ARs encountered on the potsherds and whether these can be reliably associated with use of the vessels. No sediment samples from the sites are used to assess potential contamination. The authors argue that the array of n-alkanols in a published dataset from peat deposits is not observed in the pottery, as would be expected if the ceramics were contaminated. However, the n-alcohol content of the crannog peat is unknown. As a more conclusive check, would the authors be able to see whether ARs are present in the exterior surfaces of the pottery vessels, perhaps using base portions that wouldn't reasonably be expected to have come into contact with cereals during use?

Overall, the authors have been very honest about this potential weakness and the balance of probability is, to my mind, in favour of endogenous lipids. Other than sample the exteriors, it is also hard to think what else they could have done if the sediment samples were not available for analysis. But I think this does need to be emphasised as a caveat in the main article and a recommendation for sampling associated sediments in future research. Notably, "Potential environmental sources of alkylresorcinols" is a header in the supplementary but not mentioned in the main article. It also transitions from being 'unlikely' to being 'ruled out' in the supplementary. Again I think conceding that this is a potential source but an 'unlikely one' for the reasons given would be more appropriate. A further point regarding the observed correlation in ARs by pot size and co-occurrence of other products could also be made as perhaps this is unlikely to be through random contamination with environmental lipids in the depositional environment.

Line 81-83 "The exceptional burial conditions of artefacts from four recently-discovered sites raised the possibility that prehistoric cereal processing and/or cooking in pots may be detected through the recovery of cereal-specific biomarkers that we have previously shown to be likely to survive only under anoxic conditions" The authors need to reference Colonese et al. where these markers were first reported in an archaeological context. It is surprising that this article is not mentioned when introducing the research field and is instead reserved for a brief note in the supplementary.

Fig 3 - it is interesting that cereals or other plant foods are not included on this plot. Although of far lower fatty acid content, it is nevertheless important to consider the effects of mixing substantial amounts of cereal derived fatty acids with animal products, taking into account differences in their relative C16:/C18:0 concentrations. Also no errors are shown on the mixing lines giving a misleading sense of certainty. The 1.2 per mill Suess correction is also inappropriate without knowing when the reference products were collected, as will have substantially increased since Friedli et al. 1986.

Line 317 - 326 (and abstract, etc) e.g. "Nonetheless, our findings clearly demonstrate that wheat was consumed at an early stage at the onset of farming within this region, which is in apparent contrast to previous studies, which have suggested that very little wheat was consumed in Atlantic Scotland throughout the Neolithic" -

Comparisons cannot be reasonably made between archaeobotanical finds of cereals and the

frequency of cereal residues in the pot. Firstly, although barley finds are greater than wheat finds in Atlantic Scotland they are not totally absent as shown in Bishop et al. and as the authors point out barley has a lower AR content than wheat so is far less likely to be detected. Secondly, no archaeobotanical evidence is available from the immediate study area and the entire EN archaeobotanical assemblage from the Western Isles is represented by just one site. It is not appropriate from an archaeological or environmental point of view to conflate the Northern and Western Isles. Thirdly, the number of pots with AR and the overall sample size is small, highly culturally conditioned and not necessarily representative of cereal consumption, noting other processing and preparation methods.

I strongly suggest this argument be removed from the paper and the reasons for the absence of barley, detected so far, is suggested as hypotheses that would need to be tested through further research. The abstract should also be adjusted accordingly.

Supplementary table - how is "Additional info" obtained, how is this related to the molecular identifications - please clarify? This column seems unnecessary.

Reviewer #2:

Remarks to the Author:

The paper focuses on preserved lipid biomarkers (plant and animal) from Neolithic vessels and other artefactual material at four crannogs (Loch an Duna, Loch Arnish, Loch Bhorgastail and Loch Langabhat) on the Isle of Lewis in Outer Hebrides. The authors suggested that the analysis of lipid biomarkers not only provides an exceptional opportunity to explore culinary practises in , but also to gain broader insights into the presence of cereals and the nature of their use in the Outer Hebrides during the Neolithic. The authors have shown that the superior sensitivity and selectivity of gas chromatography, combined with high-resolution mass spectrometry, allows the detection and identification of specific lipid biomarkers (e.g. alkylresorcinols) associated with cereal processing and preserved in pottery, together with lipids of animal origin (e.g. ruminant carcasses and milk fats).

We believe that this work is an original attempt to reconstruct local Neolithic culinary practises in the context of communal ceremonial behaviour with feasting at the crannogs 'liminal' location in the centuries around 4000 BC. The authors hypothesise that small-mouthed vessels ('Unstan'-type bowls) and the food cooked or simmered in them (as a kind of porridge/milk gruel) were brought to the crannogs from domestic settlements in the vicinity. The alkylresorcinols (C21 alkyl chain homologue or AR -21) associated with cereal processing and the dairy lipids (triacylglycerol patterns with broad distribution from C40 to C54) associated with cereal processing suggest that cereals were probably either boiled or cooked with milk in the pots. However, sequential use of these pots for dairy products and cooking of cereals is also possible.

The authors note that this research also demonstrates that biomolecular evidence for the presence of alkylresorcinols in ceramic matrices is significant at sites or in regions where traditional archaeological indicators of plant cultivation (such as charred plant macrofossils) have not survived. For example, excavated deposits at Loch Langabhat were sampled for charred plant remains, but there was no evidence of cereal or wheat cultivation in the Neolithic. On the other hand, archaeological and palaeobotanical data on cereal grain assemblages found in the wider region of Scotland during the Neolithic show a remarkable dominance of barley cultivation, particularly in this Atlantic region. They hypothesised that wheat and barley were prepared for consumption in different ways in this region, leading to different traces of past consumption in the archaeological record. Wheat, for example, may have been cooked in soups or porridges, resulting in minor incidental carbonization of the grains, while barley was dry-roasted on hot stones for consumption, a practise that often results in significant carbonization of the grains. However, we suggest they should take a look at the pollen records, which are generally well preserved in lake sediments.

A total of 59 Neolithic sherds were selected from crannogs. Cereal lipid biomarkers, i.e. alkylresorcinols were detected in a total of 16 samples, indicating widespread use of cereals. Lipid values of 16:0 and 18:0 fatty acids were determined in a total of 29 samples, i.e., more than half of the samples showed $\delta^{13}C$ values consistent with reference values for pure milk lipids.

Alkylresorcinols have also detected in sedges and in the organic matter of the peat formed from

them, but the authors rule out an environmental source for the biomarkers found in the ceramic samples deposited directly in the lake, and they believe that this represents a true food-related signal. On the other hand, archaeogenetic human aDNA analyses are known to indicate that the earliest occurrence of the lactase persistence and milk allele -13910*T is in a European Bell Beaker culture that dates to no earlier than 2450- 2140 cal BC.

Reviewer #3:

Remarks to the Author:

Review. Neolithic culinary traditions revealed by cereal, milk and meat lipids in pottery from Scottish crannogs by Simon Hammann et al.

This manuscript presents data from the analysis of lipid residues preserved in pottery vessels from Scottish crannogs (n = 59). It suggests that the authors are in a position to provide a picture of the role of pottery in these specific archaeological contexts, related to the identification of animal and plant food-related commodities.

The field of organic residues in pottery vessels is clearly one of the most promising fields in pottery studies. Although the data are really interesting, the low number of samples and the specific geographic region and archaeological contexts convert it into a very specific publication. The impact of the publication would probably be oriented to a more methodological readership. For that reason, this reviewer considers that the manuscript would be more suited to a specialised journal (i.e. *Scientific Reports*, *Journal of Archaeological Science*), although some aspects should be covered up. In addition, I have a number of general concerns about this manuscript and the specific sections.

Introduction

In lines 78-80 the authors support:

“By using a highly sensitive approach to analyse organic residues extracted from Neolithic pottery, we are able to directly detect specific molecular biomarkers of the cereals that were cooked in the vessels themselves.”

This idea is also revealed in lines 137-148, where the authors introduce a general view of the basis for the organic residue analysis in pottery. These principles have been well-published since 2008 and largely included in the protocols for pottery analysis, so this review considers that this brief review of organic residue analysis should be omitted. In this paragraph (lines 143-146), authors support

“We have recently demonstrated that through the superior sensitivity and selectivity achieved by gas chromatography coupled to high resolution mass spectrometry it is possible to confidently detect and identify specific lipid biomarkers related to cereal processing absorbed and preserved in pottery alongside animal-derived lipids”.

In this sense, it would be useful to include the first reference of molecular identification of cereal grains associated with archaeological artefacts: Colonese, A. C. et al. (2017) ‘New criteria for the molecular identification of cereal grains associated with archaeological artefacts’, *Scientific Reports*, 7(1), pp. 1–7. doi: 10.1038/s41598-017-06390-x.

At the end of the introduction, one of research questions proposed is focused on the functionality of the sites, specifically “can this shed light on the activities taking place at these enigmatic sites and their relation to Neolithic practices more generally in the Outer Hebrides and beyond?”. However, at the beginning of the introduction (lines 73-75), the interpretation of these sites related to ceremonial activities is directly mentioned. Given that the functionality of the sites is a key-question of the research, it would be advisable to avoid any reference to the ceremonial practices, or by contrast, to include all the possibilities about it.

Results

The authors present the results of 59 pottery samples (and 2 foodcrusts?), showing a sampling rate extremely low. The shortcomings of the sampling rate are emphasised by a range of other recent regional studies involving lipid residue analyses.

It should be advisable to include in the SI all the radiocarbon dates of the sites of Loch an Duna, Loch Arnish, Loch Bhorgastail and Loch Langabhat to offer all the chronological information of the sites. In the same way, if calibrated radiocarbon dates are presented calibration and software for calibration should be cited.

More information about the archaeological context (excavation, superficial survey,...), stratigraphic units and selection criteria should be included in the supplementary information. In addition, it would be advisable to include the total number of pots by site to recognise the representativeness of the sampling

Considering the information included in Supplementary Table 1, the percentage of lipid recovered by site ranged from 36.9 to 63.6%. This supposes that 49.2% of the selected sherds contained an amount of lipids above 5 µg/g.

Site	Total	N	N with lipids	%
Loch an Duna	11	7	63.64	
Loch Arnish	17	10	58.82	
Loch Bhorgastail	12	5	41.67	
Loch Langabhat	19	7	36.84	
TOTAL	59	29		49.15

This implies that the sentence "Across all four sites the average rate of samples with lipid contents in excess of 5 µg/g (determined by GC-FID) was about 56%." presents an overestimated percentage.

Identification of animal lipid sources using compound-specific $\delta^{13}\text{C}$ analysis of fatty acids.

Authors support that isotopic values of fatty acids were determined in a total of 29 samples (including one visible residue). However, in SI Table 1, isotopic values are provided for 29 pottery samples. There is no reference for the data obtained from these visible residues. These samples should be included in the SI with the archaeological information, provenance, isotopic and molecular information.

Figure 3 includes ellipses with the isotopic values for the palmitic and stearic fatty acids of marine fats. It should be advisable to include a summary table (maybe as supplementary information) of the modern reference fats and their references, including these marine fats. These values are the basis for Figure 3 and Supplementary Figure 1, so it would be advisable to include a table or summary table from modern authentic reference fats (ruminant, non-ruminant, dairy and marine oils) and their references.

Evidence for processing of cereals and other plants through analysis of specific lipid biomarkers

Authors also include a comment in line 226 about the incorporation of two visible residues, despite the fact that no more information (archaeological or chemical compounds) of these samples is found in the manuscript or the supplementary information. Following the SI Table 1, 14 pottery samples present AR as direct evidence of cereal processing.

Archaeobotanical information of the sites should be included in the manuscript. The authors propose that the use of some vessels is linked with crop processing, citing a general review of the British Islands. However, the authors do not present any archaeobotanical data in the manuscript. It would be advisable to correlate the lipid residue results with detailed archaeobotanical assessments.

Related to this aspect, it should be advisable to include if preliminary optical identification of visible foodcrust has been carried out prior to lipid extraction given the particularities of the archaeological contexts and their preservation.

SUPPLEMENTARY INFORMATION

In general, the paper is well referenced and illustrated, and the supplementary information seems both detailed and rigorous. However, information about the visible residues is missing from the SI. Supplementary Figure 4 is a bar-chart (not a histogram) showing the average percentage of different AR homologues in modern reference samples of cereals. The figure has to be properly labelled.

Supplementary Figure 6 and 7 are a bar chart (not histograms) showing the relative intensity of C20-C34 n-alcohols in lipids extracts.

Manuscript *Neolithic culinary traditions revealed by cereal, milk and meat lipids in pottery from Scottish crannogs* (NCOMMS-20-22112B)

Comments of reviewers and our **responses:**

Reviewer #1 (Remarks to the Author):

This is an interesting paper and makes a major advance in the detection, contextualisation and use of cereals in prehistory. Whilst the approach is not entirely novel, previous studies have demonstrated ARs in wetland contexts dating to the Roman period and in a prehistoric container from the alps, this is the first time it has been applied to prehistoric ceramics which are the most abundant artefact in the archaeological record for this type of research. It also addresses an important area of research concerning the establishment of agriculture in the British Isles and should serve as an important reference for future studies. While the sample size is low, the observation of cereals with vessel size and the co-occurrence with dairy fats provides new insight into cereal use during this period.

Reply: Thank you for your positive words about our paper and recognition of its important contribution. Importantly, however, we would dispute that sample size in our work is low. This work includes the analysis of nearly 60 samples from four sites from the same period and from a constrained region (Isle of Lewis on the Outer Hebrides). There are numerous previously published manuscripts with (i) comparable sample numbers (e.g. Fewlass et al., 2020 *Nature Human Behaviour* 4, 71 sherds from 4 sites) (ii) slightly higher numbers that span significantly larger chronological and spatial scales (e.g. Craig et al., 2013, *Nature* 496, 101 samples from 13 sites spanning over three millennia and the whole of Japan, a country with an area of 378,000 km² cf. the Isle of Lewis in our study, with an area of just over 2,000 km²), and (iii) indeed, significantly lower sample numbers from specific contexts, in some cases demonstrating a novel and labour-intensive methodology (Hendy et al., 2018, *Nature Communications* 9; 10 samples from Çatalhöyük) and others very specific practices (Dunne et al, 2019, *Nature* 574, three infant feeding vessels). Considering the special status of these sites (<https://www.bbc.com/travel/article/20210718-scotlands-mysterious-ancient-artificial-islands>), the constrained spatial and time range of these samples, limited number of pottery samples available for analysis (see below) and the elaborate analytical methodology used here, the sample number seems very appropriate and representative of *early Neolithic Hebridean crannogs*.

In terms of novelty: Previous work by Colonese et al (2017, *Scientific Reports* 7) described the analysis of in-situ residues of cereal grains in a Bronze Age wooden container preserved in an Alpine glacier. After microscopic identification of spelt, emmer and barley the team proceeded to detect cereal lipids *in morphologically-identifiable cereal residues*, but clearly the context of this find was very exceptional and this method can't be applied in many other circumstances. However, we have previously established that a) these cereal lipids can be transferred into a ceramic vessel fabric during cooking b) that diagnostic markers can be preserved under favourable circumstances as *non-visible absorbed residues* and c) they can be extracted and detected in a low number of Romano-British samples, which were analysed to demonstrate the applicability of

the methodology developed (Hammann and Cramp, 2018, Journal of Archaeological Science 93). After this method development work, we now describe in this manuscript for the first time the application of these new methods, and show that cereal lipids are absorbed and preserved in ceramic containers and can be used to infer cereal processing as far back as the British Neolithic, when pottery was first introduced to this region. We would highlight that the only earlier direct molecular evidence for cereals in prehistory comes from proteomics analysis of pottery from Çatalhöyük (Hendy et al 2018), from which 10 sherds were analysed.

The methodology is sound and builds upon previous work by the authors which has validated the approach on experimental material. One concern is the low quantity of ARs encountered on the potsherds and whether these can be reliably associated with use of the vessels. No sediment samples from the sites are used to assess potential contamination. The authors argue that the array of n-alkanols in a published dataset from peat deposits is not observed in the pottery, as would be expected if the ceramics were contaminated. However, the n-alcohol content of the crannog peat is unknown. As a more conclusive check, would the authors be able to see whether ARs are present in the exterior surfaces of the pottery vessels, perhaps using base portions that wouldn't reasonably be expected to have come into contact with cereals during use?

Reply: As the reviewer saw in the supplementary discussion we were very aware of possible environmental sources and made every effort to rule this out. We did follow the reviewer's suggestion and re-sampled the inside and outside surfaces from the three samples from Loch Bhorgastail, in which cereal biomarkers were detected (i.e. BHO16-9, BHO16-26a and BHO16-26b). Alkylresorcinols were not detected in any of the surface samples, neither inside nor outside. If the source of the alkylresorcinols were to be environmental, highest concentrations would be expected in the outside portions of the sherd.

Fortuitously, excavations resumed during the summer 2021 at Loch Bhorgastail, which allowed us to take this one step further, and collect additional environmental samples and compare the lipid compositions of the samples directly with the respective sediments. Three sediment samples from the loch bed (for locations, see Supplementary Figure 6) were collected and lipids extracted and analysed using the same methodology as for the ceramic samples. All three sediment samples contained detectable quantities of plant sterols and alkylresorcinols, but in very different compositions as were observed in the samples. The new information has been added to the manuscript and the discussion on potential environmental sources of alkylresorcinols in the supplementary information has been significantly expanded and now reads (**Lines 152-287, Supplementary Information**):

Alkylresorcinols as potential environmental signals

Since ARs can be found in sedges and sedge-derived peats^{49,50}, it is essential to explore, and discount, a possible environmental origin for the ARs identified in the pottery here. We conducted a series of experiments, including analysis of lipid-extracted sediments from the loch bed, as well as additional analyses of the inner and outer surfaces of sherds.

We believe that an environmental signature is highly unlikely as the source of the ARs we identified in the crannog pottery, due to the following conditions that could be reasonably expected for this explanation, **not** being met:

- 1) **ARs would need to be relatively mobile in watery environments.** Due to their hydrophobic structure ARs are poorly water soluble, and our own experiments showed that even in boiling water only low quantities of alkylresorcinols were liberated from the cereal grains and dissolved in water⁹. This suggests that the mobility of alkylresorcinols in such a water environment is very limited.

- 2) **ARs would be detected on (and indeed, likely concentrated on) the outer surfaces of the sherd.** If the source for the alkylresorcinols in our samples were to be the sediments or the water body itself, a passive absorption of ARs would be expected with highest concentration of ARs found on the outer parts of the sherd. On the other hand, it is well known from reference experiments that cooking of lipid-rich resources can lead to high quantities of lipids being absorbed into the inner parts of the ceramics, with highest concentrations potentially towards the inside wall of the pot⁵¹. We tested this hypothesis here by the analysis of the lipids extracted from the surfaces of three sherds from Loch Bhorgastail, which already showed detectable quantities of ARs in the absorbed residues. In contrast *ARs were not detected on the exterior (inner, nor outer) surfaces of any of these sherds.*

- 3) **ARs would either be present in every sherd, or show spatial patterning associated with burial at a particular discrete location.** In case of an environmental source of ARs, they would be expected to be present in all samples from a given site, or show spatial patterning within a site, given the close temporal and spatial association between the ceramic samples.

However, ARs were in fact only present in approximately 30% of sherds analysed and there is no spatial clustering in the locations of these sherds containing ARs (**Supplementary Figure 6**). Furthermore, ARs were present in only some samples and not others found within the same context and thus very close spatial proximity.

- Ceramic samples
- ★ Ceramic samples with cereal biomarkers
- Sediment samples
- Current loch level
- Bathymetric contours

Supplementary Figure 6: Maps of Loch Bhorgastail and Loch Langabhat showing the positions of ceramic samples analysed (yellow circles), samples with detectable quantities of cereal biomarkers (yellow stars) and sediment samples collected at Loch Bhorgastail (red squares). Please note, that some positions indicate more than one sample found within the same context.

4) Patterns of plant-derived biomarkers would be homogenous and not bear any patterning with ancient anthropogenic activity. Similar to alkylresorcinols, other environmentally derived lipid compounds would be expected to be absorbed in the ceramic sherds. In this case, for a given site a homogenous distribution of these compounds would be expected, as all were exposed to the same environmental conditions. Testing this for long chain alcohols and fatty acids, it is apparent that this is not the case in our samples where we have stark differences in terms of predominating alcohol homologue and general pattern (**Supplementary Figure 10 and 11**).

Furthermore, we observed a clear correlation of ARs with smaller vessels, as well as a co-occurrence with dairy lipids (see Figures 5 and 6). This means that their presence in pots is not random but follows a material culturally-associated pattern, and, therefore followed a specific pattern of use which is not explainable through environmental uptake.

This suggests that these cereal lipids were not absorbed through the burial environment but reflect different resource input into the individual pots and thus a clear anthropogenic signal.

5) Finger-prints of plant-derived biomarkers in pottery residues would match those in burial sediments. Ongoing excavations at Loch Bhorgastail allowed us to collect several sediment samples at this site to compare the composition of lipid biomarkers with our samples from this site, in which alkylresorcinols were detected, i.e. BHO16-9, BHO16-26a and BHO16-26b.

In all three sediment samples alkylresorcinols and plant sterols could be detected, with samples 1 and 2 only showing very low abundant peaks of AR-21 and AR-27 while sample 3 showed a wider distribution of ARs. In all three sediment samples long chain alkylresorcinol homologues (AR-25 or longer) were detected, which were completely absent in our samples (**Supplementary Figure 7+8**). These would be expected to be also present in the archaeological residues in case of an environmental input, but were not. Sediment sample 3 was

vastly more rich in alkylresorcinols and other organic material, but was collected in deeper parts of the loch bed compared to the other two sediment samples and the ceramic samples (Supplementary Figure 6). Noteworthy, in one ceramic sample found in depths similar to sediment sample 3 no cereal biomarkers were detected, thus showing no relationship between lipid composition of sediments and ceramic samples.

Supplementary Figure 7: Partial GC-QToF MS extracted ion chromatograms (m/z 268.1315) of the three sediment samples from Loch Bhorgastail as well as the AR-22 standard showing the elution of alkylresorcinols. Sediment sample 3 is shown in full scale and sediment samples 1 and 2 and the AR-22 standard are zoomed into to ca. 1% relative intensity. Small inserts on the right side show the full scale chromatogram. Please note, that a low contamination of AR-21 was found in the neat AR-22 standard (ca. 0.05%), which is **not responsible for the presence of AR-21 in the samples, where a significantly lower quantity was used. Peaks marked in bold were **not** detected in any ceramic samples.**

Supplementary Figure 8: Partial GC-QToF MS extracted ion chromatograms (m/z 268.1315, smoothed) of extracts of absorbed residues from samples BHO16-26a, BHO16-26b, and BHO16-9a showing the elution of alkyresorcinols AR-21 and AR-23 as well as the internal standard (AR-22)

Furthermore, the patterns of n-alcohols were inconsistent between extracts from the ceramic samples and the sediment samples: While in the ceramic samples C20 or C22 alcohols were most abundant, highest abundances were found at C26 or C28 in the sediment samples (**Supplementary Figure 9**). *This again speaks against a general uptake of environmental lipids into the ceramics.*

Supplementary Figure 9: Bar charts showing the distributions of C₂₀-C₃₄ *n*-alcohols in lipid extracts from sherds that contained alkylresorcinols and three sediment samples from Loch Bhorgastail (BHO), determined through GC-QToF MS analysis. The bar charts are normalized to the most abundant homologue in each sample.

6) Finger-prints of plant-derived biomarkers in pottery residues would match published compositions in peat and sedges.

Alkylresorcinols (and plant sterols) have also been detected in sedges and organic matter of peat formed from these^{49,50}. In the plants AR-21 was found to be the most abundant homologue (55% of total ARs) with further contributions of AR-23 (20%), AR-19 (16%) and AR-25 (7%). In the peat either AR-21 or AR-19 was found as the most abundant alkylresorcinol homologue depending on sample depth⁵⁰, while our pottery samples consistently featured AR-21 as main homologue and AR-19 and AR-23 were only detectable in a low number of pottery lipid extracts.

In the sedge-derived peat *n*-alkanols from C₂₀ to C₃₄ were reported with generally low abundant C₂₀ homologue and a predominating C₂₂ homologue, though this depended on the sample depth⁴⁹. In contrast, in the majority of our ceramic samples which contained ARs the C₂₀ alcohol had a relative intensity of at least 20% and indeed, in seven samples it was the most abundant alcohol (**Supplementary Figure 10**). Furthermore, in extracts from our pottery samples the C₃₂ and C₃₄ alcohols were only present in traces or were not detected at all, while these were found to be abundant in peat⁴⁹. Similarly, carboxylic acids in peat were reported to maximise at C₂₄ or C₂₆⁴⁹. In our samples, long chain fatty acids were generally only low abundant compared to the C₁₆ and C₁₈ fatty acids (**Supplementary Figure 11**). While this is common for mixtures of

degraded animal fats with plant-derived lipids there was no obvious bimodal distribution maximising both at C₁₆/C₁₈ and C₂₄/C₂₆ which would indicate significant input of plant-derived fatty acids

Therefore, AR and particularly n-alcohol and fatty acid patterns identified in the pottery residues do not match the fingerprints identified in published compositions of sedges and sedge derived peats.

With these observations in mind we therefore consider an environmental source for the ARs found in our samples highly unlikely, but the evidence strongly suggests that this represents a genuine food-related signal.

Supplementary Figure 10: Bar charts showing the distributions of C₂₀-C₃₄ *n*-alcohols in lipid extracts from sherds that contained alkylresorcinols. Samples from Loch Bhorgastail (BHO), Loch Langabhat (LAN), Loch an Duna (LAD) and Loch Arnish (LAR) are displayed in blue, black, green and red, respectively, determined through GC-QToF MS analysis. The bar charts are normalized to the most abundant homologue in each sample.

Supplementary Figure 11: Bar charts showing the distributions of C₁₂-C₃₀ fatty acids in lipid extracts from sherds which contained alkylresorcinols. Samples from Loch Bhorgastail (BHO), Loch Langabhat (LAN), Loch an Duna (LAD) and Loch Arnish (LAR) are displayed in blue, black, green and red, respectively, determined through GC-QToF MS analysis. The bar charts are normalized to the most abundant homologue in each sample. Only samples could be taken into account which were analysed by GC-MS without prior SPE enrichment step (see Table 1), since in this step the free fatty acids were separated and discarded.

We believe that this new data adds further evidence that these compounds are genuine, use-related and not environmental signals.

n-Alkanols: The author is correct, that the *n*-alkanol content of any (hypothetical) peat is not known. However, we should reiterate that the samples *were not found deposited in peat* but the silty loch beds. Our caution derives from the fact that we cannot rule out the presence of a peat over the last 5000 years, or the incursion of water flowing from peats in the environment. We write in the supplementary information (**Lines 155-161, Supplementary Information**):

“While our samples were not found deposited in peat, possible peat incursion cannot be ruled out over the course of the last 5500 years (since there is considerable run-off of water into these lochs from the local peat-rich landscape on each), which raises the question of whether an uptake of ARs into the sherds might have occurred. We therefore conducted a series of experiments, including analysis of lipid-extracted sediments from the loch bed, as well as additional analyses of the inner and outer surfaces of sherds:”

There is unfortunately no peat to analyse and provide comparison with our pottery lipid signatures. However, we rationalise that if the alkylresorcinols in our samples were an environmental signal, other related compounds, such as *n*-alkanols would be environmentally-derived as well. Given that the samples at a given site were deposited in very close temporal and spatial proximity, the environmental conditions would be the same for all of them, leading to the same or, at least, similar patterns across all of the pottery recovered from that location. However, this is not what we found at all, which is why we argue that the *n*-alkanol pattern is inconsistent with an environmental source, even without knowing the content or composition of any hypothetical source for most of the sites (See above).

In addition, the new data from Loch Bhorgastail shows a clear mismatch of *n*-alkanol patterns between the samples and the sediment (not peat) samples taken at this site. This was added to the Supplementary Information:

“In this case, for a given site a homogenous distribution of these compounds would be expected, as all were exposed to the same environmental conditions. Testing this for long chain alcohols and fatty acids, it is apparent that this is not the case in our samples where we have stark differences in terms of predominating alcohol homologue and general pattern (Supplementary Figures 10 and 11).” (**Lines 204-208, Supplementary Information**)

“Furthermore, the patterns of *n*-alcohols were inconsistent between extracts from the ceramic samples and the sediment samples: While in the ceramic samples C₂₀ or C₂₂ alcohols were most abundant, highest abundances were found at C₂₆ or C₂₈ in the sediment samples. *This again*

speaks against a general uptake of environmental lipids into the ceramics.” (Lines 252-256, Supplementary Information)

Overall, the authors have been very honest about this potential weakness and the balance of probability is, to my mind, in favour of endogenous lipids. Other than sample the exteriors, it is also hard to think what else they could have done if the sediment samples were not available for analysis. But I think this does need to be emphasised as a caveat in the main article and a recommendation for sampling associated sediments in future research. Notably, “Potential environmental sources of alkylresorcinols” is a header in the supplementary but not mentioned in the main article. It also transitions from being ‘unlikely’ to being ‘ruled out’ in the supplementary. Again I think conceding that this is a potential source but an ‘unlikely one’ for the reasons given would be more appropriate. A further point regarding the observed correlation in ARs by pot size and co-occurrence of other products could also be made as perhaps this is unlikely to be through random contamination with environmental lipids in the depositional environment.

Reply: We are glad that the reviewer appreciates our efforts to be transparent about possible other sources. As explained in the reply to the reviewer’s comment above, we have sampled the inside and outside surfaces of selected samples as well as sediment samples at one site. The results further strengthen the argument for the cereal lipids to be endogenous, use-related signals and not explainable through environmental uptake.

We have added a reference in the main text to refer to the Supplementary Discussion, also taking the reviewer’s excellent point on the correlation with pot-size and other contents into account:

“Considering the low quantities of these compounds, distinguishing genuine food-related and environmental sources is challenging, but the specific association of the cereal biomarkers with certain vessel shapes and contents, as well as the patterns of other lipid biomarkers, makes an environmental source very unlikely (see Supplementary Discussion)”. (Lines 253-257)

The following paragraph was added to the Supplementary Discussion:

“Furthermore, we observed a clear correlation of ARs with smaller vessels, as well as a co-occurrence with dairy lipids (see Figures 5 and 6). This means that their presence in pots is not random but follows a material culturally-associated pattern, and, therefore followed a specific pattern of use which is not explainable through environmental uptake.

This suggests that these cereal lipids were not absorbed through the burial environment but reflect different resource input into the individual pots and thus a clear anthropogenic signal.“

(Lines 209-215, Supplementary Information)

We also changed the last sentence of the Supplementary Discussion and removed the “ruled out”. It now reads:

“With these observations in mind we therefore consider an environmental source for the ARs found in our samples highly unlikely, but the evidence strongly suggests that this represents a genuine food-related signal. “ **(Lines 339-341, Supplementary Information)**

Line 81-83 “The exceptional burial conditions of artefacts from four recently-discovered sites raised the possibility that prehistoric cereal processing and/or cooking in pots may be detected through the recovery of cereal-specific biomarkers that we have previously shown to be likely to survive only under anoxic conditions” The authors need to reference Colonese et al. where these markers were first reported in an archaeological context. It is surprising that this article is not mentioned when introducing the research field and is instead reserved for a brief note in the supplementary.

Reply: The reviewer is of course correct that Colonese et al. reported alkylresorcinols for the first time in archaeological contexts. However, as mentioned above, their sample were very special circumstances as it was in-situ organic residues where cereals could even be identified microscopically. It is very unlikely to find such conditions at other archaeological excavations.

We have followed the reviewer’s suggestion and we modified the introduction accordingly to include a reference of Colonese et al’s work. It now reads:

“However, plants in general have previously been difficult to detect with this approach, and cereals in particular were invisible. In 2017 Colonese et al. were able to detect cereal-specific lipids extracted from the amorphous, in-situ preserved residues of a Bronze Age wooden container from Switzerland. We have recently demonstrated that these compounds can be absorbed and preserved in the matrix of ceramic vessels and it is also possible, using highly sensitive and selective methods of gas chromatography coupled to high resolution mass spectrometry, to confidently detect and identify traces of these in pottery samples alongside animal-derived lipids” **(Lines 155-162)**

Fig 3 - it is interesting that cereals or other plant foods are not included on this plot. Although of far lower fatty acid content, it is nevertheless important to consider the effects of mixing substantial amounts of cereal derived fatty acids with animal products, taking into account differences in their relative C16:/C18:0 concentrations.

Reply: We know from our own reference analysis that cereals are generally lower in fat than animal products, with GC-amenable lipids in the 1-2% range with exception of oats (See Table 1 below). However, even more pronounced than the generally lower lipid content is the extremely poorly extraction of cereals lipids during cooking. In our reference experiments the total lipid contents of experimental pots, which had been used for ten cooking steps with cereals, were in the range of 10 µg/g, while similar experiments using milk and pork led to lipid contents of ca. 0.6 and 4 mg/g, respectively, i.e. 60 to 400 times higher (Hammann et al., 2018, Journal of Archaeological Science 93).

Furthermore, we observed that alkylresorcinols were more resilient against degradation than fatty acids/acyl lipids (see Hammann and Cramp, 2018, Journal of Archaeological Science 93) this is also born out in this study, where five samples contained detectable quantities of ARs, but no interpretable quantities of fatty acids surviving alongside them. Considering the low quantities of ARs, the general lower lipid contents and extraction efficiency of cereal lipids, the contribution of cereal fatty acids to the major fatty acids will be negligible.

Table 1: Average, median and range of lipid contents of the cereals investigated (as determined by GC-FID) in mg/g dry matter (Taken from Hammann et al., 2019, Food Chemistry 282, Supporting Information).

Cereal (no of samples)	Average lipid content (GC-FID) [mg/g]	Median lipid content (GC-FID) [mg/g]	Range [mg/g]
Bread wheat (n=15)	10.0	9.6	6.9-12.8
Spelt (n=10)	12.5	12.2	10.2-14.9
Einkorn (n=11)	16.3	16.9	13.2-19.1
Emmer (n=14)	10.9	10.4	8.6-14.6
Barley (n=11)	7.0	7.2	4.9-8.6
Rye (n=8)	8.4	7.9	7.6-10.4
Oats (n=8)	38.4	38.2	24.7-52.6

Also no errors are shown on the mixing lines giving a misleading sense of certainty.

Reply: The dashed mixing lines are for orientation only and should not be seen as quantitatively interpretable feature. We specified this in the figure caption and added the following statement:

“Lines connecting the ellipses represent the calculated theoretical $\delta^{13}\text{C}$ values obtained through the mixing of these fats and are for orientation only.”

The 1.2 per mill Suess correction is also inappropriate without knowing when the reference products were collected, as will have substantially increased since Friedli et al. 1986.

Reply: We understand the reviewer's point, but believe the correction is still appropriate. The terrestrial reference ellipses present the data by Copley et al, 2003 PNAS 100 (16 ruminant adipose, 10 ruminant dairy fats, eight porcine/nonruminant adipose fats), which stems from reference samples collected during the 1990s. The correction of 1.2 ‰ has been chosen to account for the depletion of $\delta^{13}\text{C}$ of atmospheric CO_2 from the 1800s till the mid 1980s due to intensive fossil fuel burning, as described in Friedli et al or Rubino et al., Journal of Geophysical Research: Atmospheres 118). With the instrumental error associated with isotopic analysis in the region of 0.3 per mil, this will exceed any minor error. We do agree that this value will need adjustment for more recent or future reference collections.

Line 317 - 326 (and abstract, etc) e.g. “Nonetheless, our findings clearly demonstrate that wheat was consumed at an early stage at the onset of farming within this region, which is in apparent contrast to previous studies, which have suggested that very little wheat was consumed in Atlantic Scotland throughout the Neolithic” -

Comparisons cannot be reasonably made between archaeobotanical finds of cereals and the frequency of cereal residues in the pot. Firstly, although barley finds are greater than wheat finds in Atlantic Scotland they are not totally absent as shown in Bishop et al. and as the authors point out barley has a lower AR content than wheat so is far less likely to be detected.

Reply: We are not trying to say that barley was absent, but that our data indicates a processing of wheat, which was not expected from the (scarce) archaeobotanical data and extreme rarity of evidence for wheat in the Neolithic of the study region. The archaeobotanical data and AR data provide complementary approaches which together strengthen our interpretations of past food consumption practices. We are not disputing the archaeobotanical evidence but highlight that our biomarker-based approach might provide evidence for resources (or ways of processing) that are not well represented by archaeobotany. Vice versa, as the reviewer points out, barley will likely not be detectable using our biomarker approach, but is well preserved in the archaeobotanical record. All approaches have their inherent biases and can't provide a full picture of all commodities, but using several approaches and methodologies in combinations allows us to get the most accurate and comprehensive picture of past resource use. We therefore think it is correct and important to compare both approaches, keeping their biases, strengths and drawbacks in mind. To clarify this, we have added this point to the final sentence of the paragraph referred to:

“Therefore using this combined approach, with complementary data from both archaeobotany and AR significantly strengthens our interpretations about early food consumption practices.” (Lines 366-368)

We agree with the reviewer that wheat was not totally absent in Atlantic Scotland, but as discussed in our manuscript, the very small quantities of wheat recovered from Neolithic samples from the Outer Hebrides (and elsewhere in Atlantic Scotland) suggest these grains were probably weeds within the barley crop rather than evidence of deliberate wheat cultivation. See lines 347-349: “While both emmer wheat and free-threshing wheat grains have been identified in Neolithic samples from the Outer Hebrides, the small quantities so far recovered have suggested that wheat was a weed contaminant of the barley crop rather than a deliberately cultivated crop.” This is also discussed in Bishop et al 2009.

To clarify this further, we have edited the following text to state that barley as well as wheat were consumed in the region:

“Nonetheless, our findings clearly demonstrate that wheat was consumed at an early stage at the onset of farming within this region, which is in apparent contrast to previous studies, which have suggested that very little wheat was consumed in Atlantic Scotland throughout the Neolithic ⁴”

This section now reads,

“Nonetheless, our findings clearly demonstrate that wheat was consumed at an early stage at the onset of farming within this region - as well as barley - as indicated by the wider archaeobotanical data from across the region ⁴. Our data is therefore in apparent contrast to the archaeobotanical record, which suggests that very little wheat was consumed in Atlantic Scotland throughout the Neolithic” (**Lines 353-357**)

Secondly, no archaeobotanical evidence is available from the immediate study area and the entire EN archaeobotanical assemblage from the Western Isles is represented by just one site. It is not appropriate from an archaeological or environmental point of view to conflate the Northern and Western Isles.

Reply: The reviewer is right that there is currently little archaeobotanical data available for our immediate study area and the wider region for this period. However, this scarcity of available archaeobotanical data highlights the value of our complimentary biomarker-based approach that enables us to get snapshots of foodways and resource use where archaeobotanical data is unavailable, e.g., due to poor preservation. The Northern and Western Isles vary archaeologically and environmentally, but are similar in terms of climate and environmental adaptations for cereal growth and we believe it is therefore appropriate to compare our AR data to this wider archaeobotanical data as it is the best available archaeobotanical data at present to compare to our AR data (of course, more Neolithic archaeobotanical data may become available from this region in future, with more excavations and archaeobotanical analyses, but further excavations and sampling are beyond the scope of the present study). Moreover, we have stated that our data

is “in **apparent contrast** to the archaeobotanical record” to highlight that there is a degree of uncertainty. (Line 355)

Thirdly, the number of pots with AR and the overall sample size is small, highly culturally conditioned and not necessarily representative of cereal consumption, noting other processing and preparation methods.

I strongly suggest this argument be removed from the paper and the reasons for the absence of barley, detected so far, is suggested as hypotheses that would need to be tested through further research. The abstract should also be adjusted accordingly.

Reply: We understand the reviewer’s reservation but we want to emphasise that we do not in any way suggest the absence of barley but rather the (additional?) presence of wheat in ceramic vessels based on our biomarker results. The suggestion that wheat might have been present to a larger degree than previously thought (with possible reasons why this has not been reflected in the available record so far) is a very interesting result in its own right; whether this is restricted to ceremonial/’special’ sites or is more widespread in domestic contexts as well, is an exciting avenue of future research.

We fully agree that further research is needed, to identify how different use patterns would make the detection of certain cereal species more or less likely in the archaeobotanical record or in biomarker analysis. Our study here represents a first snapshot, that archaeobotanical sampling might not have identified the full range of crops cultivated during the Neolithic, and we aim to continue along this line of research.

We believe the abstract is appropriate as it highlights these thoughts and merely states that our data indicates the new evidence for the consumption of wheat and does not claim the absence of barley. We do not believe it is useful to discuss this in more detail in the abstract.

Supplementary table - how is “Additional info” obtained, how is this related to the molecular identifications - please clarify? This column seems unnecessary.

Reply: We changed the table layout: In addition to a column providing the interpretation from $\delta^{13}\text{C}$ analysis all other molecular information is summed up in the column “Other biomarkers present“

Reviewer #2 (Remarks to the Author):

The paper focuses on preserved lipid biomarkers (plant and animal) from Neolithic vessels and other artefactual material at four crannogs (Loch an Duna, Loch Arnish, Loch Bhorgastail and Loch Langabhat) on the Isle of Lewis in Outer Hebrides. The authors suggested that the analysis of lipid biomarkers not only provides an exceptional opportunity to explore culinary practises in , but also to gain broader insights into the presence of cereals and the nature of their use in the Outer Hebrides during the Neolithic. The authors have shown that the superior sensitivity and selectivity of gas chromatography, combined with high-resolution mass spectrometry, allows the detection and identification of specific lipid biomarkers (e.g. alkylresorcinols) associated with cereal processing and preserved in pottery, together with lipids of animal origin (e.g. ruminant carcasses and milk fats).

We believe that this work is an original attempt to reconstruct local Neolithic culinary practises in the context of communal ceremonial behaviour with feasting at the crannogs 'liminal' location in the centuries around 4000 BC. The authors hypothesise that small-mouthed vessels ('Unstan'-type bowls) and the food cooked or simmered in them (as a kind of porridge/milk gruel) were brought to the crannogs from domestic settlements in the vicinity. The alkylresorcinols (C21 alkyl chain homologue or AR -21) associated with cereal processing and the dairy lipids (triacylglycerol patterns with broad distribution from C40 to C54) associated with cereal processing suggest that cereals were probably either boiled or cooked with milk in the pots. However, sequential use of these pots for dairy products and cooking of cereals is also possible.

The authors note that this research also demonstrates that biomolecular evidence for the presence of alkylresorcinols in ceramic matrices is significant at sites or in regions where traditional archaeological indicators of plant cultivation (such as charred plant macrofossils) have not survived. For example, excavated deposits at Loch Langabhat were sampled for charred plant remains, but there was no evidence of cereal or wheat cultivation in the Neolithic. On the other hand, archaeological and palaeobotanical data on cereal grain assemblages found in the wider region of Scotland during the Neolithic show a remarkable dominance of barley cultivation, particularly in this Atlantic region. They hypothesised that wheat and barley were prepared for consumption in different ways in this region, leading to different traces of past consumption in the archaeological record. Wheat, for example, may have been cooked in soups or porridges, resulting in minor incidental carbonization of the grains, while barley was dry-roasted on hot stones for consumption, a practise that often results in significant carbonization of the grains. However, we suggest they should take a look at the pollen records, which are generally well preserved in lake sediments.

Reply: We are aware of the available pollen evidence from the region and we do not believe it is relevant to include this data in our paper. Cereal pollen generally does not travel far (both *Hordeum* and *Triticum* are self-pollinated) and is underrepresented in pollen records. Furthermore, the identification of cereals to genus level is problematic using pollen, because *Hordeum* and *Triticum* pollen overlap in their characteristics, as well as with wild grasses

(Andersen, 1979, Danmarks Geologiske Undersøgelse; Joly et al., 2007, Review of Palaeobotany and Palynology 146), and also cereal pollen is not always identified to genus during pollen analysis. For the isle of Lewis, the published pollen sequences do not provide identifications to genus for cereal pollen (See Fossitt, 1996; New Phytologist 132; Birks and Madsen, 1979, Journal of Ecology 67; Bohncke et al 2016). For example, the pollen analysis from peat deposits (local pollen source area) at Calanais on Lewis provides evidence for cereal cultivation in the early Neolithic in this area of Lewis, but the cereal pollen was not identified to genus (Bohncke et al, 2016, in Calanais Survey and Excavation 1979–88). The unpublished PhD thesis of Tim Lomax (1997, University of Birmingham, Holocene Vegetation History and Human Impact in Western Lewis, Scotland) provides identifications of *Hordeum* sp. pollen of later prehistoric date (Bronze Age and later) and also Mesolithic (wild *Hordeum* sp.) from several sites on Lewis, but not *Triticum* sp. pollen. It is also difficult to precisely date cereal pollen data to relate it to archaeological record, whereas it is possible to directly radiocarbon date cereal macrofossils. Therefore, the macrofossil data provides more secure data for examining relative importance of *Hordeum* and *Triticum* in the Neolithic than pollen data and the pollen data is not relevant for the purposes of our discussion.

A total of 59 Neolithic sherds were selected from crannogs. Cereal lipid biomarkers, i.e. alkylresorcinols were detected in a total of 16 samples, indicating widespread use of cereals. Lipid values of 16:0 and 18:0 fatty acids were determined in a total of 29 samples, i.e., more than half of the samples showed $\delta^{13}C$ values consistent with reference values for pure milk lipids.

Alkylresorcinols have also detected in sedges and in the organic matter of the peat formed from them, but the authors rule out an environmental source for the biomarkers found in the ceramic samples deposited directly in the lake, and they believe that this represents a true food-related signal.

On the other hand, archaeogenetic human aDNA analyses are known to indicate that the earliest occurrence of the lactase persistence and milk allele -13910*T is in a European Bell Beaker culture that dates to no earlier than 2450- 2140 cal BC.

Reply: The author is correct that dairying is widespread in Neolithic Britain and predates the occurrence and proliferation of lactase persistence, with processing to cheese representing a way to make it easier digestible (see e.g. Itan et al, 2009, PLoS Computational Biology 5, Evershed et al., 2008, Nature 455, Salque et al., 2013, Nature 493). There's also good evidence suggesting that even non-lactase-persistent population would have been able to consume milk, to a degree, and the additional food source outweighed the troubles from being unable to digest the lactose (Evershed et al, 2022, Nature forthcoming).

Reviewer #3 (Remarks to the Author):

Review. Neolithic culinary traditions revealed by cereal, milk and meat lipids in pottery from Scottish crannogs by Simon Hammann et al.

This manuscript presents data from the analysis of lipid residues preserved in pottery vessels from Scottish crannogs (n = 59). It suggests that the authors are in a position to provide a picture of the role of pottery in these specific archaeological contexts, related to the identification of animal and plant food-related commodities.

The field of organic residues in pottery vessels is clearly one of the most promising fields in pottery studies. Although the data are really interesting, the low number of samples and the specific geographic region and archaeological contexts convert it into a very specific publication. The impact of the publication would probably be oriented to a more methodological readership. For that reason, this reviewer considers that the manuscript would be more suited to a specialised journal (i.e. Scientific Reports, Journal of Archaeological Science), although some aspects should be covered up. In addition, I have a number of general concerns about this manuscript and the specific sections.

Reply: We thank the reviewer for their interest in our study, but disagree with some points of their assessment:

Journal fit: We do not think that “Scientific Reports” would be a more a specialized journal compared to Nature Communications. Also, while this manuscript is based on a novel methodology we focus the paper on the results not the technicalities of the method. We do agree that the study region here is limited, but both the methodological advance and the results have implications extending over the study region and this specific study and thus will be of high interest to the wide and multidisciplinary readership of this journal.

Sample number: Given the extensive and time-consuming laboratory procedures used in this study and the special context of the sites the sample number is appropriate, also compared to other studies (some also published in this journal such as Hendy et al, 2018, Nature Communications 9). Please see also our reply to the reviewer’s comment below

Lastly, we do not understand what should be “covered up”, and we are unable to respond to this.

Introduction

In lines 78-80 the authors support:

“By using a highly sensitive approach to analyse organic residues extracted from Neolithic pottery, we are able to directly detect specific molecular biomarkers of the cereals that were cooked in the vessels themselves.”

This idea is also revealed in lines 137-148, where the authors introduce a general view of

the basis for the organic residue analysis in pottery. These principles have been well-published since 2008 and largely included in the protocols for pottery analysis, so this review considers that this brief review of organic residue analysis should be omitted. In this paragraph (lines 143-146), authors support

Reply: As this is not a specialized journal we believe that this paragraph is beneficial for readers not entirely familiar with organic residue and, more specifically, archaeological lipid analysis. We also want to emphasise that this paper is based on different instrumental approach allowing different data to be collected compared to the usually employed low resolution single quadrupole GC-MS approaches. We have rephrased to section to reflect this (see also next comment”

“We have recently demonstrated that these compounds can be absorbed and preserved in the matrix of ceramic vessels and it is also possible, using highly sensitive and selective methods of gas chromatography coupled to high resolution mass spectrometry, to confidently detect and identify traces of these in pottery samples alongside animal-derived lipids ⁹”. (Lines 157-162)

In this sense, it would be useful to include the first reference of molecular identification of cereal grains associated with archaeological artefacts: Colonese, A. C. et al. (2017) ‘New criteria for the molecular identification of cereal grains associated with archaeological artefacts’, Scientific Reports, 7(1), pp. 1–7. doi: 10.1038/s41598-017-06390-x.

Reply: We agree, and we have added a reference. The section now reads:

“In 2017, Colonese et al. were able to detect cereal-specific lipids extracted from the amorphous, in-situ preserved residues of a Bronze Age wooden container from Switzerland” (Lines 156-157)

At the end of the introduction, one of research questions proposed is focused on the functionality of the sites, specifically “can this shed light on the activities taking place at these enigmatic sites and their relation to Neolithic practices more generally in the Outer Hebrides and beyond?”. However, at the beginning of the introduction (lines 73-75), the interpretation of these sites related to ceremonial activities is directly mentioned. Given that the functionality of the sites is a key-question of the research, it would be advisable to avoid any reference to the ceremonial practices, or by contrast, to include all the possibilities about it.

Reply: We thank the reviewer for this comment, and we fully agree. We have removed the mention of ceremonial practices in the introduction

Results

The authors present the results of 59 pottery samples (and 2 foodcrusts?), showing a sampling rate extremely low. The shortcomings of the sampling rate are emphasised by a range of other recent regional studies involving lipid residue analyses.

Reply: We disagree with the reviewer on this point for several reasons.

Firstly, we do not agree that the sampling rate is extremely low. In fact, the 59 samples are from four sites close together in time of space. It could be argued that this is more appropriate than for example looking at 101 residues from 13 sites spanning 3.2 millennia (Craig et al, 2013, Nature 496), 81 samples spanning at least three millennia (Dunne et al, 2012, Nature 486) or 246 samples from 24 sites (Cubas et al, 2020, Nature Communications 11), providing fewer data points per study site than in our study.

As for recent publications published in high impact journals, Fewlass et al analysed 71 samples from four sites (Fewlass et al. , 2020, Nature Human Behaviour 4), Dunne et al. analysed 110 samples from two sites but spanning four millennia (Dunne et al., 2017, Nature Plants 3) and Lucquin et al. analysed 143 samples spanning a 9000 year sequence (Lucquin et al, 2016, PNAS 113). Furthermore, previous publications in this journal were based on significantly lower sample numbers (e.g. Hendy et al, 2018, Nature Communications 9, 10 ceramic samples), also owing to extensive methodologies used for the analysis of these.

Also, while some studies such as the study by Cubas et al mentioned above or Debono Spiteri et al, 2016, PNAS 113, 567 samples) were based on significantly higher sample numbers (as well as investigated sites), they usually focused to observe large scale transitions and shifts while this study does not aim to explain the complete Neolithic transition in Scotland, but looks at these four sites on the Isle of Lewis. Also, these studies almost exclusively were based on firmly established methods, allowing higher throughput of samples.

Lastly, we were limited in terms of available material: These recently discovered sites have not yielded the high quantities of pottery as other, longer excavated sites and not all samples were available for destructive analysis, and justified interests of curators to preserve and archive this material from these important and exciting sites must also be considered (see also below).

Considering previous publications, the special status of these sites (<https://www.bbc.com/travel/article/20210718-scotlands-mysterious-ancient-artificial-islands>), the constrained spatial and time range of these samples, and limited number of pottery samples available for analysis and the elaborate analytical methodology used here makes the sample number appropriate and representative of *early Neolithic Hebridean crannogs*. We do not agree with the reviewer's assessment of an "extremely low sampling rate"

It should be advisable to include in the SI all the radiocarbon dates of the sites of Loch an Duna, Loch Arnish, Loch Bhorgastail and Loch Langabhat to offer all the chronological information of the sites. In the same way, if calibrated radiocarbon dates are presented calibration and software for calibration should be cited.

Reply:

We have added the radiocarbon data previously published in Garrow et al, 2017, Proc Prehist Soc 83 and Garrow and Sturt, 2019, Antiquity 93 to the supplementary information as Supplementary Table 1:

Supplementary Table 1: **Radiocarbon dates** obtained so far from Loch Arnish, Loch an Duna (Ranish) Loch Bhorgastail and Loch Langabhat. Radiocarbon ages calibrated to the calendar timescale using OxCal 4.3.2¹². Date ranges calibrated using the IntCal13 atmospheric calibration curve¹³. Note the previous publications of the OxA-coded dates¹⁴ and SUERC-coded dates¹⁵

Lab code	Site name	Material	Species/type	Context	Radiocarbon age	$\delta^{13}\text{C}$ (‰) (error \pm 0.2 per mille)	Calibrated date range (cal BC) at 95% confidence
OxA-28953	Loch Arnish	Charred residue	Internal food residue on 'Hebridean' vessel	Unstratified - loch bed	4620 \pm 30	-26.3	3510-3350
OxA-28955	Lochan Duna (Ranish)	Charred residue	Internal food residue on 'Hebridean' vessel	Unstratified - loch bed	4658 \pm 30	-26.3	3520-3370
OxA-28954	Loch Bhorgastail	Charred residue	Internal food residue on 'Hebridean' vessel	Unstratified - loch bed	4749 \pm 30	-21.6	3640-3380
SUERC-77427	Loch Bhorgastail	Wood	Salix sp. - outer rings	[30] - worked timbers east of islet	4737 \pm 24	-27.3	3630-3380
SUERC-77428	Loch Bhorgastail	Wood	Salix sp. - outer rings	[31] - worked timbers east of islet	4629 \pm 23	-27.4	3500-3360
SUERC-77434	Loch Langabhat	Charred residue	Internal food residue on 'Hebridean' vessel	Unstratified - loch bed, findspot [8]	4708 \pm 25	-26.4	3630-3380
SUERC-77432	Loch Langabhat	Wood charcoal	Alnus cf glut - small-medium branch	[52] - occupation deposits inside structure F1	3089 \pm 24	-25.4	1420-1290
SUERC-77433	Loch Langabhat	Wood charcoal	Alnus cf glut - medium branch	[57] - occupation deposits inside structure F1	2996 \pm 24	-26.2	1370-1130

More information about the archaeological context (excavation, superficial survey,...), stratigraphic units and selection criteria should be included in the supplementary information. In addition, it would be advisable to include the total number of pots by site to recognise the representativeness of the sampling

Reply: Information about the fieldwork, contexts and assemblages were published in interim reports (see References 15 and 17), which are available online here:

<https://crannogs.soton.ac.uk/dissemination/publications-interim-reports/>

We refer to this in the Materials and Methods section:

“All of this material is described in detail within recent interim reports ^{15,17}” (Lines 458-459).

Of the available Neolithic material, between and 20% (12/59, Loch Bhorgastail) and 61% (11/18, Loch an Duna) per site was analysed (see table below). In sample selection, rim sherds were prioritised, as these are generally expected to provide highest lipid yields (Charters et al, 1997, Journal of Archaeological Science 24). Further limitations in sampling were present from the prerequisite that only sherds of a certain size and shape can be reasonably sampled and extracted.

Lastly, given that this is a destructive analysis, the interests of museums must be considered as well, and many of the larger and decorated sherds were deemed too valuable for destructive sampling.

Table 2: Overview over recovered vessels, number of Neolithic vessels and number of analysed samples for the four sites

Site	Approximate No of vessels	No of vessels from Early/Middle Neolithic	Number of sherds analysed
Loch an Duna	69	18	11
Loch Arnish	79	58	17
Loch Bhorgastail	59	59	12
Loch Langabhat	81	81	19

All this information is freely available to any reader, who is particularly interested in this aspect, under the references mentioned above. We therefore do not see any benefit of adding this to the manuscript.

Considering the information included in Supplementary Table 1, the percentage of lipid recovered by site ranged from 36.9 to 63.6%. This supposes that 49.2% of the selected sherds contained an amount of lipids above 5 µg/g.

Site Total N N with lipids %
 Loch an Duna 11 7 63.64
 Loch Arnish 17 10 58.82
 Loch Bhorgastail 12 5 41.67
 Loch Langabhat 19 7 36.84
 TOTAL 59 29
 49.15

This implies that the sentence “Across all four sites the average rate of samples with lipid contents in excess of 5 µg/g (determined by GC-FID) was about 56%.” presents an overestimated percentage.

Reply: We do not quite understand the reviewer’s numbers above, but believe they might have counted the samples with GC-C-IRMS data instead of samples with lipid contents above 5 µg/g as “positive? Actually, considering the lipid content (as written in the manuscript) the recovery rate is 58% (34/59).

Some samples did contain quantifiable lipid contents but did not yield enough of C16:0/C18:0 fatty acid for reliable GC-C-IRMS since other fatty acids or other lipids such as ketones (as in sample BHO16-13-1a) contributed significantly to the total lipid content.

Table 3: Samples analysed (A), samples yielding lipids in excess of 5 µg/g (B) and calculated recovery rate (B/A) per site

	Ceramic samples analysed	Samples with lipid content > 5 µg/g	Recovery rate
Loch an Duna	11	7	64%

Loch Arnish	17	12	71%
Loch Bhorgastail	12	7	58%
Loch Langabhat	19	8	42%
Total	59	34	58%

Identification of animal lipid sources using compound-specific $\delta^{13}\text{C}$ analysis of fatty acids.

Authors support that isotopic values of fatty acids were determined in a total of 29 samples (including one visible residue). However, in SI Table 1, isotopic values are provided for 29 pottery samples. There is no reference for the data obtained from these visible residues. These samples should be included in the SI with the archaeological information, provenance, isotopic and molecular information.

Reply: The results from analyses of the visible residues of samples LAR15-43 and LAR19-52 are listed in Supplementary Table 2 together with the other results

Figure 3 includes ellipses with the isotopic values for the palmitic and stearic fatty acids of marine fats. It should be advisable to include a summary table (maybe as supplementary information) of the modern reference fats and their references, including these marine fats. These values are the basis for Figure 3 and Supplementary Figure 1, so it would be advisable to include a table or summary table from modern authentic reference fats (ruminant, non-ruminant, dairy and marine oils) and their references.

Reply: The data for the aquatic reference ellipses has been published before (Cramp and Evershed, 2014, *Treatise in Geochemistry* 12(2)). The terrestrial reference fats are as per the existing reference (27?). The figure captions for Figure 3 and Supplementary Figure 1 were modified to reflect this. They now read:

“All $\delta^{13}\text{C}$ values obtained for modern reference terrestrial animal fats were from animals raised on pure C_3 diet²⁷ and have been adjusted for the post-Industrial Revolution effects of fossil fuel burning, by the addition of 1.2 ‰³¹. Reference values for aquatic fats are from marine species from UK waters, Atlantic and North Sea (marine) and a UK lake and Kazakh river (freshwater fish)³²”

Evidence for processing of cereals and other plants through analysis of specific lipid biomarkers

Authors also include a comment in line 226 about the incorporation of two visible residues,

despite the fact that no more information (archaeological or chemical compounds) of these samples is found in the manuscript or the supplementary information. Following the SI Table 1, 14 pottery samples present AR as direct evidence of cereal processing.

Reply: The results from chemical analysis are in Supplementary Table 2 together with the ceramic samples.

Archaeobotanical information of the sites should be included in the manuscript. The authors propose that the use of some vessels is linked with crop processing, citing a general review of the British Islands. However, the authors do not present any archaeobotanical data in the manuscript. It would be advisable to correlate the lipid residue results with detailed archaeobotanical assessments.

Reply: There are only very limited archaeobotanical assemblages available for these sites, which makes our lipid data even more valuable. We have modified a sentence in the results section to emphasise the lack of archaeobotanical data: We also included new data coming from the recent excavation at Loch Bhorgastail

“Only very limited number of cereal macrofossils (none from the Neolithic) or other archaeobotanical remains have been recovered from Loch Langabhat and Loch Bhorgastail so far (none for the other two sites, which have not been excavated yet), but extensive analysis of archaeobotanical data from across Scotland reveals that the available *Triticum* sp. crops in this period include emmer wheat (*T. turgidum* L. ssp. *dicoccum* (Schrank) Thell.) and free-threshing wheat (*T. aestivum* ssp. *aestivum*/ *T. turgidum* L. ssp. *durum*/ *T. turgidum* L. ssp. *turgidum*), with emmer the dominant wheat crop”⁴ (Lines 275-280)

“Excavated deposits at Loch Langabhat were sampled for charred plant remains but the single cereal grain recovered was of indeterminate species and was associated with Middle Bronze Age occupation deposits ¹⁵. Ongoing analysis of newly excavated material from Loch Bhorgastail has so far yielded only two grains identified as barley, and one as unspecified cereal (unpublished data).” (Lines 340-344)

Related to this aspect, it should be advisable to include if preliminary optical identification of visible foodcrust has been carried out prior to lipid extraction given the particularities of the archaeological contexts and their preservation.

Reply: This study did not encompass microscopic analysis of food crusts

SUPPLEMENTARY INFORMATION

In general, the paper is well referenced and illustrated, and the supplementary information seems both detailed and rigorous. However, information about the visible residues is missing from the SI.

Reply: We did not conduct microscopic analysis the visible residues, therefore no further information can be provided. The results from the chemical analyses of the visible residues of samples LAR15-43 and LAR19-52 are listed in table S1

Supplementary Figure 4 is a bar-chart (not a histogram) showing the average percentage of different AR homologues in modern reference samples of cereals. The figure has to be properly labelled.

Supplementary Figure 6 and 7 are a bar chart (not histograms) showing the relative intensity of C20-C34 n-alcohols in lipids extracts.

Reply: Thank you, corrected as suggested.

Reviewers' Comments:

Reviewer #1:

Remarks to the Author:

Sample size: I agree the sample is representative of "early Neolithic Hebridean crannogs" the question is whether it is large enough to comment more broadly on the 'establishment of agriculture in Neolithic Britain'. However, as I mentioned in my first review, I think this is a useful step towards the larger goal and does not require further comment or alteration from the authors.

Contamination: Excellent to see the additional analyses which I hope improves the paper and raises the bar for studies of this kind. No further comments.

Isotopes and mixing: Whilst the amount of fatty acids in cereals is far lower it is entirely conceivable that the proportion of cereals was 60 to 400 by weight greater than animal products. For example, 1kg of dry wheat mixed with 100mL (~10g dry weight) of milk. This is pertinent, as mixing with cereals would have a greater effect on the C16:0 d13C value of the mixture than the C18:0 potentially altering the offsets that are used to assign animal products. As there is no indication from the plots that the isotope values are substantially moved to a C3 cereals range are observed here, I don't think this is a problem but the authors should acknowledge that such effects may be possible.

Suess correction: Provided only values from the 1990s are used then I think the corrections are appropriate. For future reference, more recently collected authentic reference fats could be easily corrected using numerous published curves (see ref below). Similar curves also exist for marine systems including mixing and this is standard practice in the field.

Hellevang, H., Aagaard, P. Constraints on natural global atmospheric CO₂ fluxes from 1860 to 2010 using a simplified explicit forward model. *Sci Rep* 5, 17352 (2015).
<https://doi.org/10.1038/srep17352>

Wheat vs Barley: The change to 'apparent' is noted however I still feel that the interpretation of this data is overstated and still misleading given the difference in AR content between the crops which surely will have a major influence on the relative identification of these products in pottery. Simply put, if the observed frequency of barley ARs were 10x that of wheat derived ARs would that still be 'surprising and significant'? Wheat is present in Scottish EN botanical assemblages, it is of unknown relative importance in the Western Isles and likely over-represented in pottery due to its AR content. If the point above can be addressed more substantively, with additional caveats and perhaps some suggestions for future research, I am happy to recommend this excellent paper for publication.

Reviewer #2:

Remarks to the Author:

The revised manuscript (including the authors' responses to the reviewers' comments and the appended supplementary materials) is well written, methodologically sound, and a highly interesting contribution, so I recommend that it be accepted for publication as is.

Reviewer #3:

Remarks to the Author:

Dear authors,

First of all, I would like to thank you for considering all our comments in the review of your manuscript. I have carefully read all your modifications and I consider that it is a really good contribution for the organic residue analysis field.

Looking forward to reading your research,

M.

Manuscript *Neolithic culinary traditions revealed by cereal, milk and meat lipids in pottery from Scottish crannogs* (NCOMMS-20-22112B)
– Second revision

Comments of reviewers and our **responses:**

Reviewer 1 (Remarks to the Author):

Sample size: I agree the sample is representative of “early Neolithic Hebridean crannogs” the question is whether it is large enough to comment more broadly on the ‘establishment of agriculture in Neolithic Britain’. However, as I mentioned in my first review, I think this is a useful step towards the larger goal and does not require further comment or alteration from the authors.

Reply: Thank you, no amendments required

Contamination: Excellent to see the additional analyses which I hope improves the paper and raises the bar for studies of this kind. No further comments.

Reply: Thank you, no amendments required

Isotopes and mixing: Whilst the amount of fatty acids in cereals is far lower it is entirely conceivable that the proportion of cereals was 60 to 400 by weight greater than animal products. For example, 1kg of dry wheat mixed with 100mL (~10g dry weight) of milk. This is pertinent, as mixing with cereals would have a greater effect on the C16:0 $\delta^{13}\text{C}$ value of the mixture than the C18:0 potentially altering the offsets that are used to assign animal products. As there is no indication from the plots that the isotope values are substantially moved to a C3 cereals range are observed here, I don’t think this is a problem but the authors should acknowledge that such effects may be possible.

Reply: We have added a paragraph to the discussing the possibility and interpretative consequences of skewed isotope values from cereal fatty acids, and why we don’t believe that’s the case here. The paragraph reads:

‘Noteworthy, cereal lipids also contain the fatty acid 16:0 and a relatively low proportion of 18:0. If a substantial proportion of cereals were prepared in pottery alongside low quantities of meat, this would draw the value of the 16:0 fatty acid to exhibit a more depleted $\delta^{13}\text{C}$ value²³. Given the relatively low quantity of C_{18:0} fatty acid in lipids, in this scenario the C_{18:0} fatty acid would rather retain a predominantly animal-derived signature, thus reducing the $\Delta^{13}\text{C}$ value and potentially masking a dairy fat contribution. Given that the $\delta^{13}\text{C}$ values of these residues are not notably depleted, and dairy fats are widely identified, we do not think this scenario plausible in this instance’. (Lines 418-424)

Suess correction: Provided only values from the 1990s are used then I think the corrections are appropriate. For future reference, more recently collected authentic reference fats could be easily corrected using numerous published curves (see ref below). Similar curves also exist for marine systems including mixing and this is standard practice in the field.

Hellevang, H., Aagaard, P. Constraints on natural global atmospheric CO₂ fluxes from 1860 to 2010 using a simplified explicit forward model. *Sci Rep* 5,

17352 (2015). <https://doi.org/10.1038/srep17352>

Reply: We appreciate the reference provided by the reviewer, no amendments required here.

Wheat vs Barley: The change to 'apparent' is noted however I still feel that the interpretation of this data is overstated and still misleading given the difference in AR content between the crops which surely will have a major influence on the relative identification of these products in pottery. Simply put, if the observed frequency of barley ARs were 10x that of wheat derived ARs would that still be 'surprising and significant'? Wheat is present in Scottish EN botanical assemblages, it is of unknown relative importance in the Western Isles and likely over-represented in pottery due to its AR content. If the point above can be addressed more substantively, with additional caveats and perhaps some suggestions for future research, I am happy to recommend this excellent paper for publication.

Reply: We amended the discussion chapter to emphasise again the possibility of mixtures of barley and wheat, and that our findings do not mean that barley was absent.

"In fact, our approach appears to have rendered substances visible that have in the past not been identified using traditional modes of analysis"

Was changed to

"As mentioned above, given the tenfold higher abundance of ARs in wheat compared with barley, these findings do not preclude the processing of barley alongside wheat in these pots. However, as the current role of wheat versus barley is currently unknown on the Neolithic Outer Hebrides in comparison with wider Atlantic Scotland, our approach appears to have rendered substances visible that have in the past not been identified using traditional modes of analysis." (Lines 390-394)

Reviewer 2 (Remarks to the Author):

The revised manuscript (including the authors' responses to the reviewer's comments and the appended supplementary materials) is well written, methodologically sound, and a highly interesting contribution, so I recommend that it be accepted for publication as is.

Reply: Thank you, no amendments required.

Reviewer 3 (Remarks to the Author):

Dear authors,

First of all, I would like to thank you for considering all our comments in the review of your manuscript. I have carefully read all your modifications and I consider that it is a really good contribution for the organic residue analysis field. Looking forward to reading your research

Reply: Thank you, no amendments required.